# Designing novel multiepitope mRNA vaccine targeting Hendra virus (HeV): An integrative approach utilizing immunoinformatics, reverse vaccinology, and molecular dynamics simulation

Ahmad Abdullah Mahdeen[1☯], Imam Hossain[1☯]*, Md. Habib Ullah Masum[2],
Sajedul Islam[1], T. M. Fazla Rabbi[1]

1 Department of Microbiology, Noakhali Science and Technology University, Noakhali, Bangladesh,
2 Faculty of Biotechnology and Genetic Engineering, Department of Genomics and Bioinformatics,
Chattogram Veterinary and Animal Sciences University (CVASU), Chattogram, Bangladesh

☯ These authors contributed equally to this work.
* ihossain.mbg@nstu.edu.bd

pone.0312239

University, REPUBLIC OF KOREA

## Abstract

Human and animal health is threatened by Hendra virus (HeV), which has few treatments.
This *in-silico* vaccine design study focuses on HeV G (glycoprotein), F (fusion protein), and
M (matrix protein). These proteins were computationally assessed for B and T-cell epitopes
after considering HeV strain conservation, immunogenicity, and antigenicity. To improve
vaccination immunogenicity, these epitopes were selectively ligated into a multiepitope con-
struct. To improve vaccination longevity and immunological response, adjuvants and linkers
were ligated. G, F, and M epitopes were used to create an mRNA HeV vaccine. Cytotoxic,
helper, and linear B-lymphocytes' epitopes are targeted by this vaccine. The population cov-
erage analysis demonstrates that multi-epitope vaccination covers 91.81 percent of CTL
and 98.55 percent of HTL epitopes worldwide. GRAVY evaluated the vaccine's well-charac-
terized physicochemical properties -0.503, indicating solubility and functional stability.
Structure analysis showed well-stabilized 2° and 3° structures in the vaccine, with alpha
helix, beta sheet, and coil structures (Ramachandran score of 88.5% and Z score of -3.44).
There was a strong affinity as shown by docking tests with TLR-4 (central score of -1139.4
KJ/mol) and TLR-2 (center score of -1277.9 KJ/mol). The coupled V-apo, V-TLR2, and V-
TLR4 complexes were tested for binding using molecular dynamics simulation where
extremely stable complexes were found. The predicted mRNA structures provided signifi-
cant stability. Codon optimization for *Escherichia. coli* synthesis allowed the vaccine to
attain a GC content of 46.83% and a CAI score of 1.0, which supports its significant expres-
sion. Immunological simulations indicated vaccine-induced innate and adaptive immune
reactions. Finally, this potential HeV vaccine needs more studies to prove its efficacy and
safety.

**Data Availability Statement:** All relevant data are within the manuscript and its Supporting Information files.

**Funding:** The author(s) received no specific funding for this work.

**Competing interests:** The author(s) declared no potential conflicts of interest with respect to the research, authorship, and/or publication of this article.

**Abbreviations:** HeV, Hendra virus; WHO, World Health Organization; NiV, Nipah virus; N, Nucleoprotein; P, Phosphoprotein; M, Matrix protein; F, Fusion protein; G, Glycoprotein; MHC, Major histocompatibility complex; TCRs, T cell receptors; CTL, Cytotoxic T lymphocyte; TH, helper T cell; Linear B-lymphocyte, linear BCL; IEDB, Immune Epitope Database; ANNs, Artificial Neural Networks; IFN-γ, Interferon-gamma; HMM, Hidden Markov Model; GRAVY, Grand average of hydropathicity; TLR-2, Toll-like receptor 2; TLR-4, Toll-like receptor 4; PDB, Protein Data Bank; MM/GBSA, Molecular mechanics with generalised Born and surface area solvation; MD, Molecular dynamic; GROMACS, GROningen MAchine for Chemical Simulations; ns, Nanosecond; RMSD, Root mean square deviation; RMSF, Root mean square fluctuation; Rg, Radius of gyration; SASA, Solvent accessible surface area; JCat, Java Codon Adaptation Tool; CAI, Codon adaptation index; COVID-19, Coronavirus Disease 2019.

# 1. Introduction

An emerging zoonotic virus known as the Hendra virus (HeV) is related to the Nipah virus through genetics. Among the *Paramyxoviridae* family of viruses is the genus *Henipavirus*. Fruit bats (*Pteropus* spp.) are the virus's principal reservoirs, but contact with infected horses can cause transmission to people [1]. HeV poses severe threats to both human and equine health, causing acute respiratory and neurological conditions. In humans, HeV infections manifest as encephalitis, influenza-like symptoms, or relapse encephalitis, often escalating to pneumonitis and multi-organ failure [2,3]. Infected horses exhibit severe respiratory distress, fever, facial edema, ataxia, and foamy nasal discharge [4,5]. The initial outbreak of HeV occurred in September 1994 in a Brisbane suburb, where an unknown virus caused severe respiratory illness, leading to the death of thirteen horses and two human infections. The virus was renamed Hendra virus after the site of the outbreak; its original name was Equine Morbil-livirus [5]. Since that time, HeV outbreaks have affected the equine population in Australia on an almost annual basis, resulting in a mortality rate of approximately 75%. The outbreaks have primarily affected horses [6]. Human infections, though less frequent, have shown a mortality rate of 57%, underscoring the virus's lethality [5]. Nucleocapsid (N), phosphoprotein (P), matrix (M), fusion (F), glycoprotein (G), and large (L) or RNA polymerase proteins are the six key structural proteins encoded by the approximately 18,234 nucleotides of single-stranded, negative-sense RNA that makes up HeV's genome. The sequence of these proteins is 3′-N-P-M-F-G-L-5′ [7]. The virus's replication, transcription, and assembly are contingent upon the presence of these proteins. The N protein handles genome packaging and replication, while the L protein facilitates RNA synthesis and mRNA processing. The phosphoprotein gene's (P gene) co-transcriptional mRNA editing produces V and W proteins, which inhibit interferon responses [3,8]. The viral receptors ephrinB2 and ephrinB3 are bound by the surface glycoproteins G and F to permit the fusion of the viral and host cell membranes, thereby infecting host cells. Through viral infection or glycoprotein transfection, syncytia, which are multinucleated large cells, are formed through this fusing process [9–11]. Additionally, M pro-teins, play a crucial role in assembling and releasing virus particles. They concentrate at cellu-lar membrane regions during virus assembly, where they attract viral glycoproteins, viral ribonucleoproteins (RNPs), and host budding machinery [3,12,13]. The G, F, and M proteins are intriguing candidates for vaccine development due to their critical roles in HeV pathogene-sis. By selecting appropriate epitopes among such proteins and integrating them with adju-vants and linkers to boost immunogenicity, it is possible to construct a chimeric multiepitope vaccine. This vaccine has the potential to offer comprehensive protection against Hendra virus (HeV) by evoking powerful immune responses to several viral components. HeV infections in humans often result in severe respiratory diseases characterized by congestion, hemorrhage, and inflammation, mimicking acute respiratory distress syndrome (ARDS) [12,13]. The virus's ability to infect and replicate in human lower respiratory epithelial cells triggers immune acti-vation and cytokine release, contributing to the inflammatory response [14–16]. HeV can also enter the central nervous system through the olfactory nerve or the hematogenous route, lead-ing to thrombosis, necrosis, vasculitis, and other neurological complications [17–19]. At this time, neither vaccination nor a treatment for HeV illness has been authorized for use in humans [20]. Despite numerous investigations into potential treatments, such as ribavirin, chloroquine, polyIC12U, and heptad peptide fusion inhibitors, results from animal models have been inconsistent. Vaccination, however, offers a promising strategy for preventing HeV infections and mitigating outbreaks. Most research has focused on soluble recombinant G gly-coprotein (HeV-sG) [2]. Since 2012, an equine anti-Hendra virus subunit vaccine has been used in Australia, and a subunit vaccine derived from the Hendra virus attachment

glycoprotein ectodomain is undergoing Phase I clinical trials for human use [2]. There is much promise in immunotherapy, especially with multi-epitope vaccines (MEVs). The overlapping CTL, HTL, and linear BCL epitopes that make up MEVs provide a powerful tool for the treatment and prevention of viral infections and cancers [21,22]. However, there are no FDA-approved mRNA vaccine for HeV. The mainstay of RNA vaccines is messenger RNA (mRNA), which is designed to promote antigen translation in APCs. Rapid degradation of messenger RNA (mRNA) vaccinations may reduce the risk of adverse effects and autoimmune illnesses, which is one argument in favor of using mRNA vaccines over DNA vaccines. And other types of vaccines such as subunit, viral like particles, killed, attenuated etc. The other advantages of mRNA vaccines are that they do not have the ability to cause cancer for not incorporating into host DNA, can be quickly and easily made and they trigger the body's natural defense mechanisms, which build an impenetrable barrier against infections by doing things like producing antibodies and cellular immunity [23,24]. Recent advances in immunoinformatics and reverse vaccinology have identified promising MEV candidates. Reverse vaccinology involves in identifying antigenic peptides or epitopes from pathogens using *in-silico* tools to assess their immunogenicity, stability, and efficacy, facilitating targeted DNA, mRNA and peptide vaccine development [25]. Computer scientists, mathematicians, chemists, biochemists, genomics experts, and proteomics researchers work together in immuno-informatics to improve computational models used in immunology research. This domain allows for comprehensive analysis of immune system functions, aiding in vaccine design, immune response prediction, and protein structure comparison [26]. Immuno-informatics accelerates the identification of immune responses, improves understanding of pathogen evolution and disease evasion, and supports public health through vaccine development and autoimmune disease research [27,28]. The goal of reverse vaccinology is to screen all potential peptides to identify antigenic epitopes, protecting vulnerable populations, including pregnant women, people of color, and the elderly, using AI and algorithms targeting specific B and T-cell epitopes. Advances in computational vaccinology and immunological informatics have significantly progressed vaccine development [25,29]. Our objective was to investigate innovative vaccine design strategies in response to the continued threat posed by the Hendra virus and the constraints of existing vaccine strategies. We developed an mRNA vaccine candidate that targets the Hendra virus by utilizing *in-silico* methodologies. The purpose of this investigation is to investigate whether an in silico-designed mRNA vaccine can effectively induce an immune response against the Hendra virus. Additionally, the study will ascertain the efficacy and safety profiles of the vaccine in comparison to current vaccine strategies. This vaccine will incorporate CTL, HTL, and linear BCL epitopes from the glycoprotein (G), fusion protein (F), and matrix protein (M), as well as appropriate adjuvants and linkers.

## 2. Materials and methods

### 2.1. Protein sequence retrieval

The full amino acid sequences of the *Henipavirus hendraense* (HeV) G, F, and M proteins were queried utilizing the NCBI protein database (https://www.ncbi.nlm.nih.gov/protein/) and stored in FASTA format for future use in research [30]. The accession numbers for these proteins are NP_047112.2, AEQ38140.1, and QYC64603.1.

### 2.2. Cytotoxic T lymphocyte (CTL) epitope prediction

Critical T cell responses to intracellular infections are dependent on CTLs. They use MHC class I molecules to bind to surface peptides on damaged cells, allowing them to be recognized [31]. Aided by the IEDB Analysis Resource (http://tools.iedb.org/mhci/) [32,33] together with

the Net-MHC 4.0 server (http://www.cbs.dtu.dk/services/NetMHC/) [34,35]. G, F, and M protein CTL binding epitopes were anticipated. The allergenicity, toxicity, and antigenic properties of the predicted epitopes were evaluated using the following servers: VaxiJen 2.0 (http://www.ddg-pharmfac.net/vaxijen/VaxiJen/VaxiJen.html), AllerTOP v.2.0 (https://www.ddg-pharmfac.net/AllerTOP/index.html), and ToxinPred (https://webs.iiitd.edu.in/raghava/toxinpred/algo.php) [36].

## 2.3. Helper T lymphocyte (HTL) epitope prediction

All three proteins—G, F, and M—had their HTL epitopes predicted using the MHC-II prediction module of the IEDB Analysis Resource (http://tools.iedb.org/mhcii/) and the NetMHCII-pan 4.0 server (http://www.cbs.dtu.dk/services/NetMHCIIpan/) [32,33]. The NetMHCIIpan 4.0 server uses Artificial Neural Networks (ANNs) to forecast the MHC-II binding epitope or HTL epitope [33,37]. The chosen epitopes were tested for their antigenic characteristics using the VaxiJen 2.0 server [38] (http://www.ddg-pharmfac.net/vaxijen/VaxiJen/VaxiJen.html), allergenicity prediction by AllerTOP v.2.0 server [36] (https://www.ddg-pharmfac.net/AllerTOP/index.html), and toxicity prediction by ToxinPred server [39] (https://webs.iiitd.edu.in/raghava/toxinpred/algo.php), respectively.

## 2.4. Linear B-lymphocyte (linear BCL) epitope prediction

Based on the antigen sequence features, we predicted the epitopes for linear B-lymphocyte of the G, F, and M proteins using the IEDB Analysis Resource (http://tools.iedb.org/bcell/) and ABCpred servers [40]. The ABCpred server is able to detect linear B-cell epitopes by combining a Hidden Markov Model (HMM) with a propensity scale methodology, as stated in [40]. We used the VaxiJen 2.0 server [38] (http://www.ddg-pharmfac.net/vaxijen/VaxiJen/VaxiJen.html), the AllerTOP v.2.0 server [36] (https://www.ddg-pharmfac.net/AllerTOP/index.html), and the ToxinPred server [39] (https://webs.iiitd.edu.in/raghava/toxinpred/algo.php) of their respective databases to learn about the antigenic properties, allergenicity and toxicity of the selected epitopes.

## 2.5. Analysis of population coverage

Regional and ethnic diversity influence the prevalence of distinct HLA alleles [41,42]. The IEDB Analysis Resource (population coverage) was applied for judging the vaccine candidate's population coverage [43] (http://tools.iedb.org/population/). The selected CTL and HTL epitopes and associated MHC alleles were used in conjunction with one another and separately for this aim. In addition, we highlighted the extensive distribution of particular alleles across different continents and all over the worlds.

## 2.6. Vaccine construct formation

The vaccine was ultimately created by incorporating all of the picked and highly regarded epitopes from the G, F, and M proteins. Proper adjuvants and specific epitopes were required to be incorporated into the exhaustive multiepitope vaccine through the use of applicable linkers. Linkers are used to separate epitopes, with the goal of ensuring that the features of flexibility, cuttability, and solidity of the epitopes are not disrupted in any way. The immunogenicity of an epitope vaccination can be improved by using AYY and EAAAK linkers [44]. In vaccines, the AK linker can keep HTL epitopes active independently of the immune system [45]. The linear BCL epitopes can be effectively linked using the KFER linker. EAAAK is a rigid peptide linker that forms α-helixes and has a closed-packed backbone, allowing for intramolecular

hydrogen bonding. In contrast to flexible linkers, rigid linkers offer numerous advantages. By maintaining a constant distance between the epitopes with minimal interference, EAAAK linkers offer an efficient separation of the functional domains, thereby preserving their individual functional properties [46]. The linkers EAAAK, AYY, AK, and KFER underwent deployment to ligate the adjuvant (50S ribosomal protein L7/L12) to the targeted epitopes in order to finalize the vaccine map.

## 2.7. Assessment of the physicochemical characteristics

The physicochemical characteristics of the designed vaccine construct were thoroughly analyzed employing the Expasy ProtParam server (http://web.expasy.org/protp aram/). There was an evaluation of the following characteristics: atomic number, molecular weight, total amino acid count, instability and aliphatic index, isoelectric point (pI), and GRAVY [47]. After that, the SOSUI server (https://harrier.nagahama-i-bio.ac.jp/sosui/mobile/) and the SOLpro server (https://scratch.proteomics.ics.uci.edu/) were served to forecast the vaccine's soluble nature [48–51]. According to the results indicated by the AllergenFP v.1.0 (http://ddg-pharmfac.net/AllergenFP/) [52], AllerTOP v.2.0 (https://www.ddg-pharmfac.net/AllerTOP/) [36], and AlgPred (https://webs.iiitd.edu.in/raghava/algpred/submission.html) server [52], the vaccine candidate is not known to cause allergic reactions. Two more sites, SCRATCH Protein Predictor (http://scratch.proteomics.ics.uci.edu [53] and VaxiJen 2.0 http://www.ddg-pharmfac.net/vaxijen/VaxiJen/VaxiJen.html) [38], were used to verify the antigenicity of the vaccine construct.

## 2.8. 2˚ (2D) structure prediction of vaccine

The vaccine construct's two-dimensional structure of the amino acid sequence was accurately predicted using PSIPRED (http://bioinf.cs.ucl.ac.uk/psipred/) (https://academic.oup.com/bioinformatics/article/16/4/404/187312) [54]. PSIPRED uses position-specific prediction analysis, namely Psi-BLAST, to identify and select sequences that have great similarity to the vaccine protein. Additionally, the two-dimensional structure of the vaccine was assessed using the GOR IV SECONDARY STRUCTURE PREDICTION METHOD (https://npsa-pbil.ibcp.fr/cgi-bin/npsa_automat.pl?page=/NPSA/npsa_gor4.html) and SOPMA SECONDARY STRUCTURE PREDICTION METHOD (https://npsa-pbil.ibcp.fr/cgi-bin/npsa_automat.pl?page=/NPSA/npsa_sopma.html) [55–58]. To estimate the two-dimensional structure of proteins, GOR4 makes use of Bayesian statistics and information theory [59]. Yet, with only the protein sequence and the three-mode representation of the 2˚ structure (alpha-helix, beta-sheet, and coil), the SOPMA service is still able to identify about 69.5% of amino acids [57].

## 2.9. Prediction, refinement, and validation of 3˚ (3D) structures

Researchers used an online tool known as I-TASSER (https://zhanggroup.org/I-TASSER/) to predict the 3˚ structure of the proposed multiepitope vaccine. I-TASSER combines threading, ab initio modeling and structural assembly, making it more versatile for a wide range of proteins, especially when homologous templates are available, whereas AlphaFold2 and Raptorx primariliy rely on deep learning models, which can sometimes struggle with proteins lacking homologs. Besides, I-TASSER is more user friendly than the other two servers and that is why we chose I-TASSER. This program offers models for repetitive template segment assembly and various threading alignments to aid in protein structure prediction [60,61]. For the purpose of developing a refined model of the vaccine, the GalaxyWEB server (https://galaxy.seoklab.org/cgi-bin/submit.cgi?type=REFINE) was employed [62]. The 3˚ model underwent evaluation using the SAVES v6.0 server, which uses a Ramachandran plot to verify the stereostructural

potency of the predicted vaccination model (https://saves.mbi.ucla.edu/) [63–66]. Following that, the Z-score, a metric for the 3˚ model's credibility, was calculated using the ProSA-web server (https://prosa.services.came.sbg.ac.at/ prosa.php) [67,68].

## 2.10. Engineering the bonds with disulfide (disulfide engineering)

In disulfide engineering, disulfide linkages are engineered to be part of protein structures. Proteins with disulfide linkages have a more stable folded shape because they raise the denatured state free energy and lower the conformational entropy [69]. We identified potential cysteine disulfide bonding pairs within the vaccine structure using the Disulfide by Design 2.13 algorithm (http://cptweb.cpt.wayne.edu/DbD2/) [70].

## 2.11. Conformational B-lymphocyte epitope prediction

Under the aid of the DiscoTope 2.0 (http://www.cbs.dtu.dk/services/DiscoTope/) and the Elli-Pro (http://tools.iedb.org/ellipro/) servers, conformational B-lymphocyte epitopes underwent prediction to have a legitimate 3˚ structure [71,72].

While the Disco Tope 2.0 server predicts the epitopes by utilizing surface accessibility computation [72]. Residue protrusion index (PI) calculation, PI value clustering of nearby residues, and ellipsoid approximation of the protein structure are the three methods used by ElliPro. These techniques are all implemented by ElliPro [72]. In the epitope prediction parameters for ElliPro, the minimum score and the maximum distance were set to 0.5 and 6, respectively.

## 2.12. Docking of the vaccine component

In an effort to evaluate the interaction between the vaccine construct and its receptors, docking between ligand (vaccine) and receptors was implemented. To serve as the receptors, TLR-2 (Toll-like receptor 2) (PDB: 2Z7X) and TLR-4 (Toll-like receptor 4) (PDB: 3FXI) underwent retrieval from the Protein Data Bank (PDB) server [54]. To examine the built vaccine's docking properties with the two receptors, the CLUSPRO 2.0 server (https://cluspro.org/home.php) was used. Pymol (https://pymol.org/) were used to display and project the docked complex.and PDBsum (http://www.ebi.ac.uk/thornton-srv/databases/pdbsum/Generate.html) was used to identify salt bridge, $H_2$ bonds interaction between vaccine and TLRs. In CLUSPRO 2.0 server, the chain id was blanked as per server requirement to use both chains [73–76].

## 2.13. Molecular mechanics with generalized Born and surface area solvation for free energy computation (MM-GBSA)

We used the MM-GBSA methods, which rely on molecular mechanics and the Generalized Born method, to determine the free energy of the interaction between the "V-TLR-2" and the "V-TLR-4". Bound and electrostatic interactions, van der Waals forces, polar and non-polar components, and the consequences of bound interactions are some of the molecular mechanics techniques that are being considered [77–79]. The polar solvation component is computed on the HawkDock server employing the Generalized Born equation [77–79].

## 2.14. Molecular dynamic simulation

The molecular dynamic (MD) simulation was employed to evaluate the stability of the "V-apo" (apo means unbound state of a protein, meaning the vaccine is in a state without binding to any receptors [80], "V-TLR-2" and "V-TLR-4" (V = vaccine) complexes in a synthetic physiological setting. The simulation experiment was conducted using the GROningen MAchine for Chemical Simulations (GROMACS) (version 2022.3) [81], and a water box was

constructed using the TIP3 water model [82] in conjunction with the CHARMM36m force field [83]. The boundaries of the water box were kept at a distance of 1 nanometer from the protein's exterior. To further neutralize the systems, the required ions were employed. With periodic boundary conditions and a two-fissure time integration phase, a molecular dynamic simulation occurred in 50 nanoseconds (ns). Isothermal-isochoric (NVT), isobaric (NPT), and energy minimization were subsequently performed on the system. The sampling interval for contour data analysis was determined to be 100 picoseconds (ps). The modules integrated within the software dubbed GROMACS performed calculations for root mean square deviation (RMSD), root mean square fluctuation (RMSF), radius of gyration (Rg), and solvent accessible surface area (SASA) after the simulation was executed.

## 2.15. Structure prediction of mRNA vaccine

The secondary structures of the mRNA of the vaccine were predicted and evaluated by the RNAfold (http://rna.tbi.univie.ac.at/cgi-bin/RNAWebSuite/RNAfold.cgi) server [84]. The server calculates the minimal free energy (MFE) of the query mRNA structures using thermodynamic principles [85,86]. The JCat optimized DNA sequences of the vaccine's were transcribed into RNA sequences using the DNA<->RNA conversion procedure available at http://biomodel.uah.es/en/lab/cybertory/analysis/trans.htm. Afterwards, the RNA sequence was utilized in the RNAfold server to predict and validate the secondary structure.

## 2.16. Optimization of codons and virtual cloning

Back-translation was carried out using the JCat tool (http://www.prodoric.de/JCat) to reverse translate the ordering of amino acid of the vaccine construct into a nucleotide sequence [31]. This server utilizing the *Escherichia Coli* strain K12 codon framework can assess the protein's expression level by determining the codon adaptation index (CAI) and the GC content [87,88]. For the purpose to clone the optimized vaccine gene sequence into the *E. coli* plasmid vector pET-28a(+), the restriction sites Eco53kI and PshAI were inserted into the N-terminal and C-terminal of the vaccine sequence, respectively. The final stage requires the use of the SnapGene software (https://www.snapgene.com/free-trial/). In order to ascertain the potential vaccine expression, the modified vaccine sequence was ligated on pET-28a(+) expression plasmid.

## 2.17. Simulating the immune response

The immune simulation analysis of the vaccine was conducted using the platform popularly known as C-ImmSim (https://kraken.iac.rm.cnr.it/C-IMMSIM/) [89]. The platform is capable of forecasting humoral and cellular-mediated immune reactions of mammalian hosts to a vaccine [90,91]. Before administering the vaccine, a regimen consisting of three doses was established, with each dose to be administered every four weeks. All simulation parameters, such as the total quantity of adjuvants and antigen administrations (100 and 1000) and the duration of the steps (one, eighty-four, and one hundred eighty-six), were initially assigned to their respective default values. If lipopolysaccharides were absent, the simulated vaccine's amount and phases were altered to 50 and 1000, respectively. By default, a randomly generated seed (12,345) was implemented.

## 3. Results

### 3.1. Protein sequence retrieval

The amino acid sequences of the G, F, and M proteins of HeV were obtained from the NCBI database. The sequences were utilized in this study to predict CTL, HTL, and linear B-cell

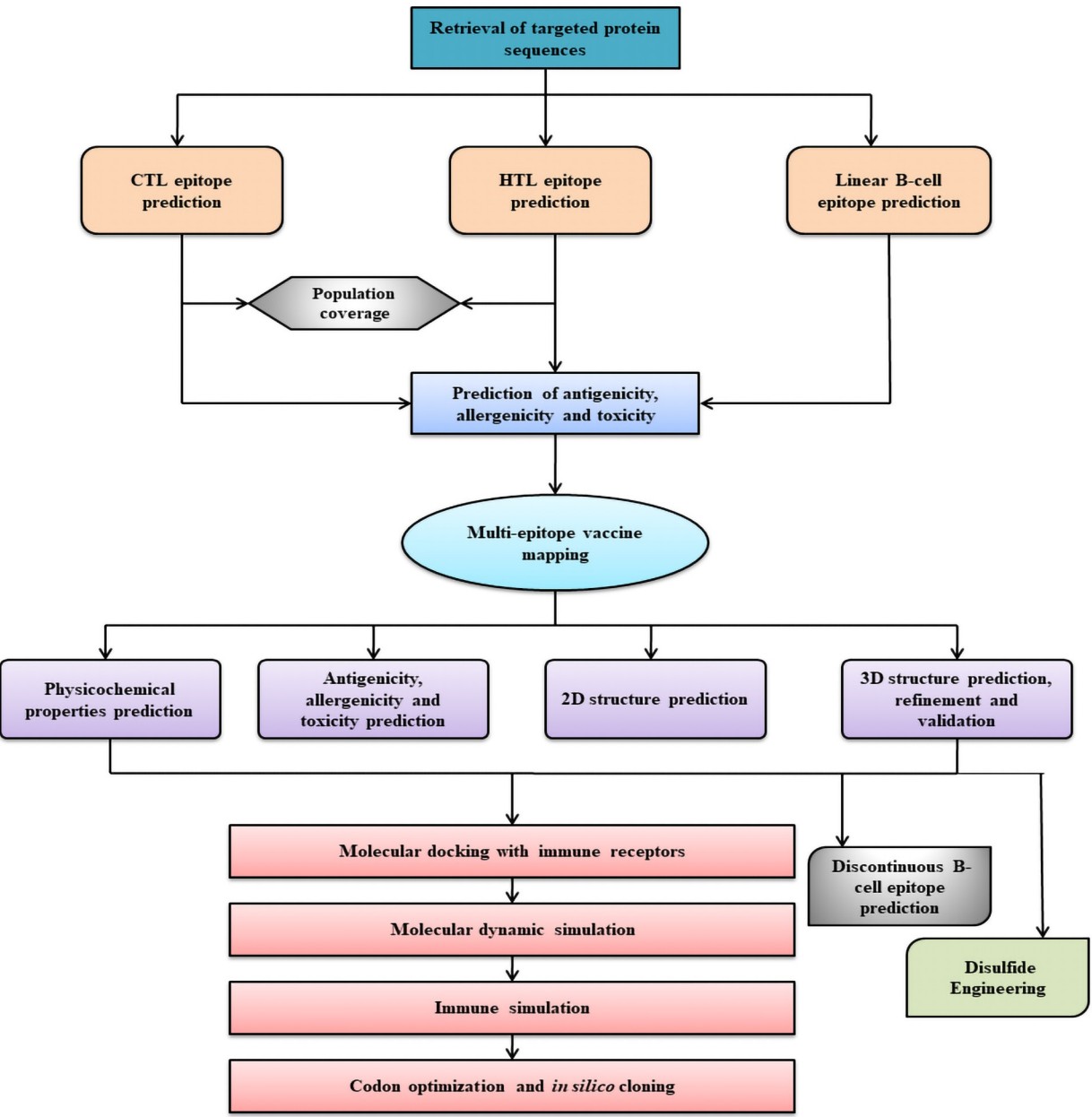

**Fig 1. A summary of the study on the development of a chimeric multiepitope mRNA vaccine.**

epitopes for the multi-epitope mRNA vaccine design. The study's plan of action is illustrated (**Fig 1**).

### 3.2. Cytotoxic T lymphocyte (CTL) epitope prediction

The IEDB database and Net-MHC 4.0 server were employed to predict all likely CTL interaction epitopes, utilizing the selected regions of the G, F, and M proteins. A selection of six peptides, each consisting of nine amino acids, was selected based on their percentile rank ($\leqq 1.00$) and affinity score, as indicated in **Table 1**. In case of CTL epitopes, the peptide presentation was restricted to the most frequent alleles in all over the world including HLA-A*01:01,

**Table 1. An inventory of chosen CTL (MHC-I) epitopes derived from the G, F, and M proteins, accompanied by their allergenic, antigenic, and cytotoxic properties.**

| Protein | Peptide | Rank of percentile | No Allele | Antigenicity | Allergenicity | Toxicity | Start | End |
|---|---|---|---|---|---|---|---|---|
| G protein | TEIGPKVSL | 0.01 | 27 | 1.4043 (likely antigen) | Potentially non-allergic | Free of toxins | 103 | 111 |
|  | TTSTILKPR | 0.01 | 27 | 0.4515 (likely antigen) | Potentially non-allergic | Free of toxins | 193 | 201 |
| F protein | KIKSNPLTK | 0.01 | 27 | 0.7250 (likely antigen) | Potentially non-allergic | Free of toxins | 47 | 55 |
|  | KLSKIGLVK | 0.02 | 27 | 0.5490 (likely antigen) | Potentially non-allergic | Free of toxins | 32 | 40 |
| M protein | VSDFSPTSW | 0.01 | 27 | 1.5702 (likely antigen) | Potentially non-allergic | Free of toxins | 16 | 24 |
|  | KVASFMLHL | 0.02 | 27 | 0.4839 (likely antigen) | Potentially non-allergic | Free of toxins | 231 | 239 |

HLA-A*02:01, HLA-A*02:03, HLA-A*02:06, HLA-A*03:01, HLA-A*11:01, HLA-A*23:01, HLA-A*24:02, HLA-A*26:01, HLA-A*30:01, HLA-A*30:02, HLA-A*31:01, HLA-A*32:01, HLA-A*33:01, HLA-A*68:01, HLA-A*68:02, HLA-B*07:02, HLA-B*08:01, HLA-B*15:01, HLA-B*35:01, HLA-B*40:01, HLA-B*44:02, HLA-B*44:03, HLA-B*51:01, HLA-B*53:01, HLA-B*57:01, HLA-B*58:01. Moreover, these peptide sequences met the requirements of being probable antigenic and free from toxicity, allergenicity prior to their use in the design of potential vaccines.

### 3.3. Helper T lymphocyte (HTL) epitope prediction

The G, F, and M proteins' HTL binding epitopes were predicted using the IEDB and NetMH-CIIpan 4.0 server. The G, F, and M proteins were used to synthesize a collection of six peptides, each of which contains 15 amino acids. These peptides were picked according to the affinity they possess and percentile classification. The frequent alleles that were chosen were HLA-DRB1*01:01, HLA-DRB1*03:01, HLA-DRB1*04:01, HLA-DRB1*04:05, HLA-DRB1*07:01, HLA-DRB1*08:02, HLA-DRB1*09:01, HLA-DRB1*11:01, HLA-DRB1*12:01, HLA-DRB1*13:02, HLA-DRB1*15:01, HLA-DRB3*01:01, HLA-DRB3*02:02, HLA-DRB4*01:01, HLA-DRB5*01:01, HLA-DQA1*05:01, HLA-DQB1*02:01, HLA-DQA1*05:01, HLA-DQB1*03:01, HLA-DQA1*03:01, HLA-DQB1*03:02, HLA-DQA1*04:01, HLA-DQB1*04:02, HLA-DQA1*01:01, HLA-DQB1*05:01, HLA-DQA1*01:02, HLA-DQB1*06:02, HLA-DPA1*02:01, HLA-DPB1*01:01, HLA-DPA1*01:03, HLA-DPB1*02:01, HLA-DPA1*01:03, HLA-DPB1*04:01, HLA-DPA1*03:01, HLA-DPB1*04:02, HLA-DPA1*02:01, HLA-DPB1*05:01, HLA-DPA1*02:01, HLA-DPB1*14:01 Additionally, the peptide sequences that were selected satisfied the criteria of being antigenic, non-allergenic, and non-toxic. Additionally, the epitopes were found to have the capacity to stimulate γ interferon in absence during IL-10 stimulation. All of the peptides that were evaluated were subsequently utilized in the production of vaccines (**Table 2**).

### 3.4. Linear B-cell epitope (linear BCL) prediction

Linear B-cell epitopes were identified by predicting potential peptide sequences from the G, F, and M proteins using the IEDB and ABCpred databases. Six linear B-lymphocyte peptide sequences were chosen for their antigenicity, non-toxicity, and non-allergic properties. Subsequently, these sequences were integrated into the vaccine formulation (**Table 3**).

### 3.5. Analysis of population coverage

The IEDB instrument for population coverage analysis was employed to determine the population coverage of specific CTL and HTL epitopes. The epitopes are capable of safeguarding an

**Table 2. Evaluation of particular HTL (MHC-II) epitopes derived from the proteins G, F, and M, along with their allergenicity, antigenicity, capacity to induce IFN- and IL-10, and toxicity.**

| Protein | Peptide | Adjusted rank | Antigenicity | Allele | Allergenicity | IFN gamma | IL10pred | Toxicity | Start | End |
|---|---|---|---|---|---|---|---|---|---|---|
| G protein | YNQKYIAITKVERGK | 0.01 | 0.9682 (likely antigen) | 27 | Potentially non-allergic | Positive | IL10 non-inducer | Free of toxins | 330 | 344 |
| | NQKYIAITKVERGKY | 0.01 | 0.9396 (likely antigen) | 27 | Potentially non-allergic | Positive | IL10 non-inducer | Free of toxins | 331 | 345 |
| F protein | ISFVIVEKKRGNYSR | 0.01 | 0.5911 (likely antigen) | 27 | Potentially non-allergic | Positive | IL10 non-inducer | Free of toxins | 513 | 527 |
| | TFISFVIVEKKRGNY | 0.01 | 1.5956 (likely antigen) | 27 | Potentially non-allergic | Positive | IL10 non-inducer | Free of toxins | 511 | 525 |
| M protein | RKKIRTIAAYPLGVG | 0.01 | 0.8600 (likely antigen) | 27 | Potentially non-allergic | Positive | IL10 non-inducer | Free of toxins | 16 | 24 |
| | KRKKIRTIAAYPLGV | 0.02 | 0.4099 (likely antigen) | 27 | Potentially non-allergic | Positive | IL10 non-inducer | Free of toxins | 231 | 239 |

estimated 91.81 (CTL) and 98.55 (HTL) percent among the world population. The population coverage for combined CTL and HTL epitopes was estimated to be 100%. The maximum population coverage (100%) is achieved in Europe through the simultaneous utilization of epitopes (Both CTL and HTL). Immediately following this are the following regions: North America (100%), East Asia (100%), North Africa (100%), South America (95.15%), East Africa (100%), Southeast Asia (99.98%), Central Africa (100%), Southwest Asia (99.99%), South Africa (95.27%), Northeast Asia (100%), Southeast Asia (99.98%), and Oceania (97.88%) (**Fig 2**).

## 3.6. Vaccine construct formation

The vaccine construct was produced by utilizing a variety of suitable epitopes of the G, F, and M proteins. During antigen presentation, peptide linkers were positioned between each epitope to facilitate the conjugation and separation of the epitopes. Consequently, a singular peptide was formed by combining CTL, HTL, and linear B-cell epitopes that were either in close proximity to one another or overlapped. The vaccine was designed to incorporate the selected adjuvant and epitopes through the use of linkers, which included EAAAK, AYY, AK, and KFER. The adjuvant was connected using the EAAAK linker (50S ribosomal protein L7/L12) to the epitopes of CTL. The CTL epitopes were connected through the use of linkers known as AK, while the HTL epitopes were connected through the use of linkers known as AYY. Furthermore, the linkers notably known as KFER were of paramount importance for connecting the epitopes of linear B-lymphocyte. Consequently, ours multiepitope vaccine candidate was developed with a single adjuvant, six CTL, six HTL, and six linear B cell epitopes, arranged in a trajectory from the N terminal to the C terminal (**Fig 3**).

**Table 3. A listing of chosen linear BCL epitopes accompanied by their allergenic, antigenic, and toxicity.**

| Protein | Start | End | Peptides | Toxicity | Allergenicity | Antigenicity |
|---|---|---|---|---|---|---|
| G protein | 377 | 391 | KSSIESTNEAVVKLQE | Free of toxins | Potentially non-allergic | 0.6356 (likely antigen) |
| | 393 | 456 | RTLGYATEDFDDLLES | Free of toxins | Potentially non-allergic | 0.4969 (likely antigen) |
| F protein | 433 | 460 | LGSINYNSESIAVGPP | Free of toxins | Potentially non-allergic | 1.2114 (likely antigen) |
| | 292 | 387 | YVQELLPVSFNNDNSE | Free of toxins | Potentially non-allergic | 0.6083 (likely antigen) |
| M protein | 63 | 106 | MYMICYGFVEDVERSP | Free of toxins | Potentially non-allergic | 0.6657 (likely antigen) |
| | 301 | 361 | | Free of toxins | Potentially non-allergic | 0.7398 (likely antigen) |

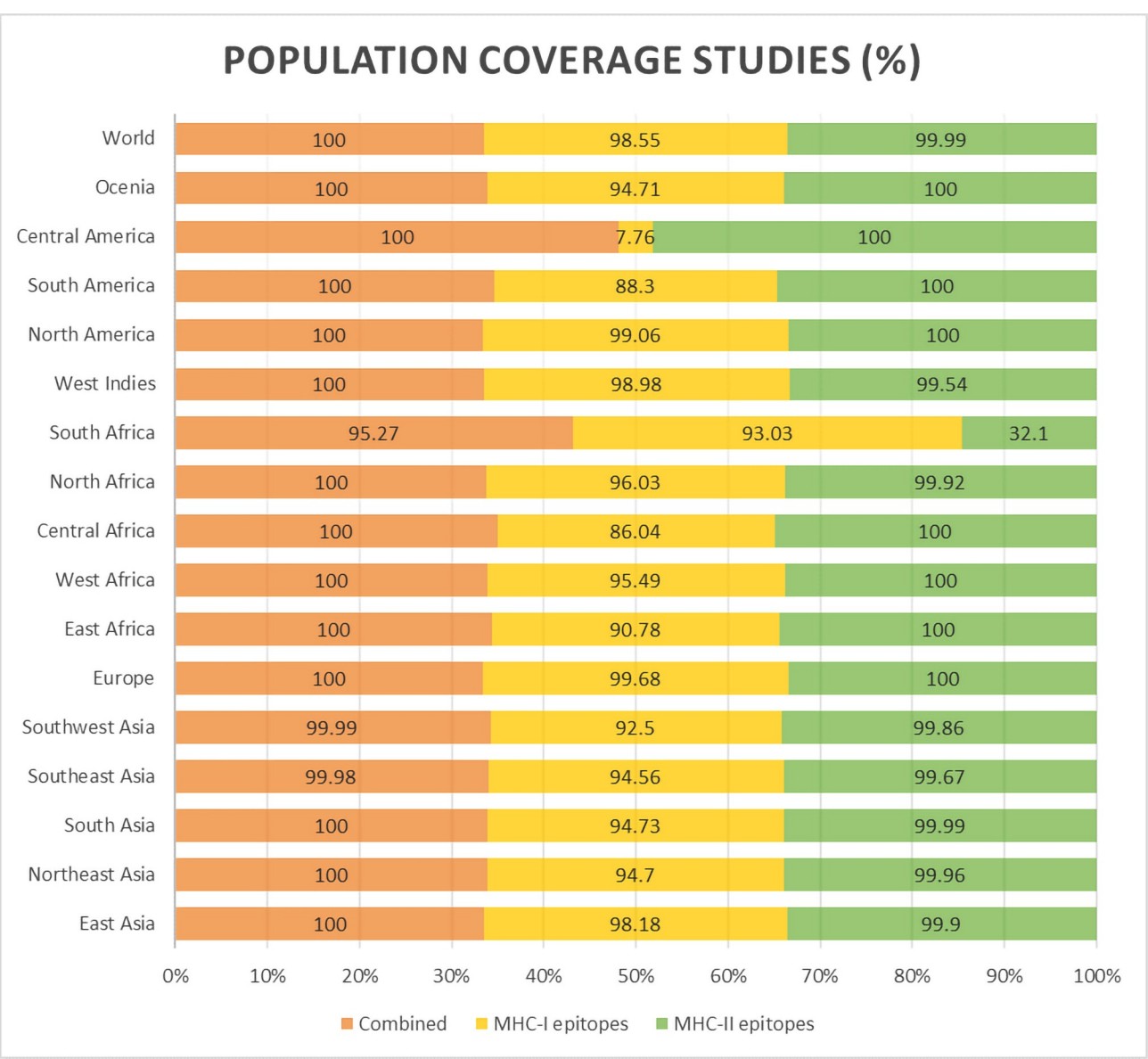

**Fig 2. Vaccine population coverage of the epitopes (CTL, HTL and combined) that qualified for inclusion in the HeV vaccine construct.**

### 3.7. Assessment of the physicochemical characteristics of the vaccine

The physicochemical characteristics of the vaccine were identified by employing the Expasy-ProtParam database (Table 4). The mass of vaccine molecule was 55349.73Da and the amino acid count was 494, as indicated by the server. The isoelectric point (pI) of 9.76 is indicative of its alkaline nature. Furthermore, the vaccine's anticipated GRAVY score of -0.308 implies that it is hydrophilic, or soluble in water, which is also predicted by a hydrophobicity score of -0.345 through the SOSUI server. The SOLpro server suggested that the vaccine had the potential to dissolve effectively when expressed in *E. coli*, as evidenced by its score of 0.859720. Nevertheless, the index of stability of 37.18 suggests that it is stable. Table 4 indicates that three popular servers AllergenFP v. 1.0, AllerTOP v. 2.0 and AlgPred servers confirmed the absence

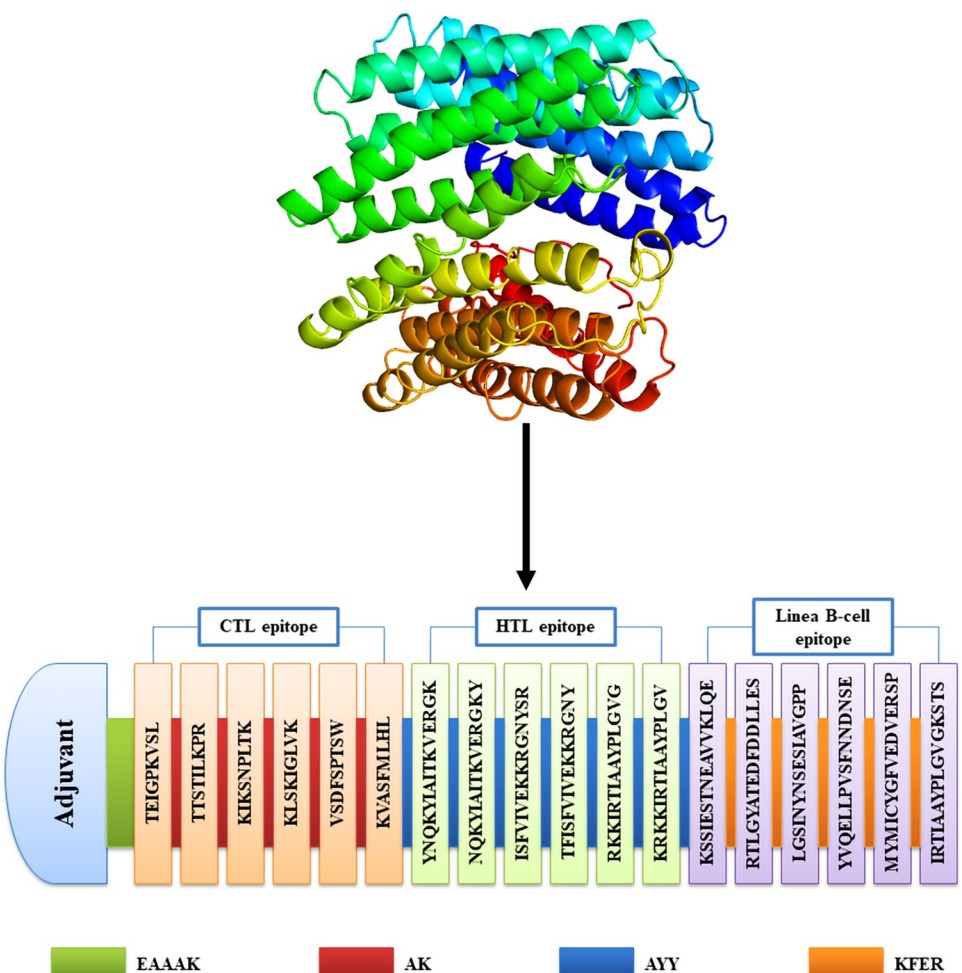

**Fig 3. The assembly of a multiepitope vaccine targeting HeV and the ribbon 3˚ model of the vaccine.** The schematic illustration of a peptide sequence consisting of 494 amino acids include adjuvant (sky blue color), epitopes (CTL-coral orange, HTL-tea green, B-cell-light violet), and linkers (EAAAK-green, AYY-blue, AK-maroon, KFER-orange).

of allergic reactions to this vaccine. The VaxiJen 2.0 and ANTIGENpro servers ultimately identified our designed vaccine as an antigen of potential qualities.

### 3.8. 2˚ (two-dimensional) structure prediction of vaccine

A combination of the GOR4, SOPMA, and PSIPRED servers was employed to forecast the vaccine's 2˚ structure. The GOR4 server contains 59.11% alpha helices, 26.92% random coils, and 13.97% extended strands (beta sheets). The SOPMA server predicted that the vaccine's 2˚ structure would consist of a random coil (23.68%), extended strands (18.22%), and an alpha helix (53.64%) (Table 5, Fig 4B and 4C). The PSIPRED server's three-state approach predicts the protein's secondary structure, which is depicted (Fig 4A). This approach includes strands, coils, and helices.

### 3.9. Prediction, refinement, and validation of 3˚ (3D) structures

The initial model (Fig 3) was selected on account of its superior TM-score of 0.49±0.15, C-score of -1.82, and RMSD of 11.6±4.5Å, out of the multiple models estimated by the server

**Table 4. Physicochemical characteristics of the HeV-vaccine.**

| Physicochemical characteristics | Findings |
|---|---|
| Mass of molecule (Da) | 55349.73 |
| Amino acid count | 494 |
| Conceptual pI | 9.76 |
| GRAVY | -0.503 |
| Index of stability | 37.18 |
| Index of the aliphatic | 82.11 |
| The total number of residues that are negatively charged (Asp+Glu) | 59 |
| The total number of residues that are positively charged (Arg+Lys) | 93 |
| Number of atoms | 7928 |
| Solubility (SOSUI/ SOLpro) | Soluble protein |
| Allergenicity (AllerTOP v. 2.0/ AllergenFP v.1.0/ AlgPred) | Potentially non-allergic |
| Antigenicity (VaxiJen 2.0/ ANTIGENpro) | Likely antigen |

known as I-TASSER. Subsequently, another server named as GalaxyWEB was employed to retrieve the 3˚ model that was improved. Due to the 88.5% Ramachandran's preferred region, 0.421 RMSD, and 2.707 MolProbity scores, the 3˚ structure showed excellent stability. The RMSD of 0.575 Å was determined by combining the GalaxyWEB refined and I-TASSER predicted models. This result confirms that the refined model is appropriate for further investigation. The energy-minimizing model allocated 88.5% of its amino acid residues to the most favored regions, 9.1% to the increased allowed regions, and 0.9% to the generously allowed regions, as evidenced by the Ramachandran plot (**Fig 5A**). The values of the preliminary model were 77.9%, 16.8%, and 3.5%, respectively. The server called ProSA confirmed that the enhanced model with a Z-score of -3.44. Conversely, the preliminary model's score of -2.69 undoubtedly suggests that the projected model is more accurate (**Fig 5C**). Because the more the negative Z score values, the better the model structure will be. The refined model has an ERRAT score of 89.834 (**Fig 5B**).

## 3.10. Disulfide engineering of vaccine

After using the Disulfide by Design 2.13 server to assign disulfide bonds, 42 pairings of amino acids were found to be eligible. When additional factors like energy and Chi3 value were

**Table 5. The GOR4 and SOPMA algorithm findings.**

| Properties | GOR4 | SOPMA |
|---|---|---|
| Alpha helix (Hh) | 292 (59.11%) | 265 (53.64%) |
| $3_{10}$helix (Gg) | 0.00% | 0.00% |
| Pi helix (Ii) | 0.00% | 0.00% |
| Beta bridge (Bb) | 0.00% | 0.00% |
| Extended strand (Ee) | 69 (13.97%) | 90 (18.22%) |
| Beta turn (Tt) | 0.00% | 35 (10.70%) |
| Bend region (Ss) | 0.00% | 0.00% |
| Random coil (Cc) | 133 (26.92%) | 117 (23.68%) |

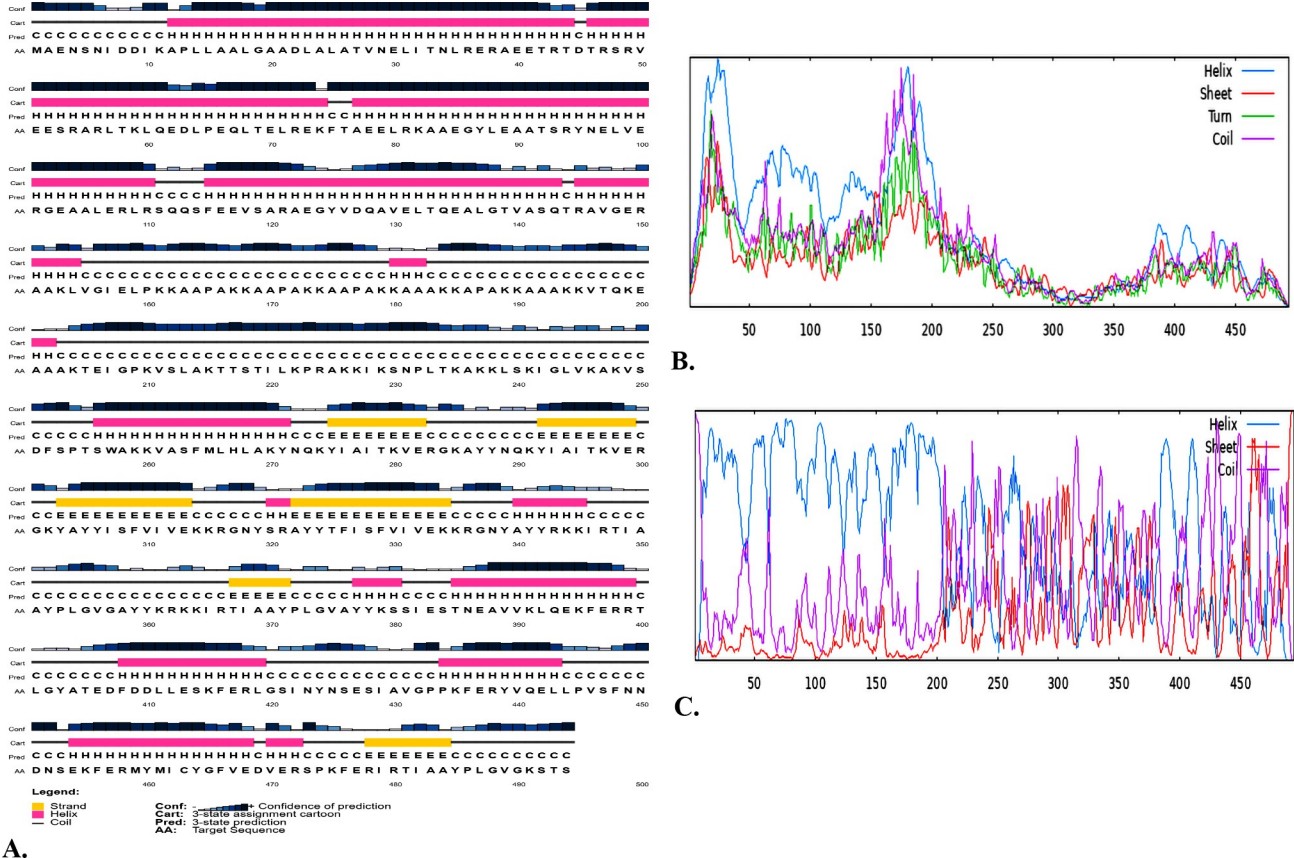

**Fig 4. 2° structure of HeV vaccine.** (A) The 2° vaccine structure that was identified via PSIPRED where confidence level in the prediction is denoted by the initial bar (Conf), and the magnitude of each subsequent bar corresponds to a distinct level of confidence. In the second bar (Cart), the beta-sheet is denoted by the yellow color, the helix by the pink color, and the vaccine's coil structure by the grey color. The third bar (Pred) and fourth bar (AA) correspond to three discrete structural elements and sequences of amino acids, respectively, (B) Graphical representation of the 2° structure of vaccine by SOPMA, (C) Presentation of the 2° structure of the vaccine in graphical form by GOR4.

considered, though, the number of couples dropped to just 2. So, five mutations were introduced at the GLU 117-ALA 120, SER 5-ASP 8, ILE 311-LYS335, GLU 412–GLY 432, and LYS 246–METT 265 pairs of residues (**S1 Fig**). Chi3 values between -87 and +97 and energy values below 2.2 were suggested for disulfide engineering. These couples are listed (**S1 Table**).

## 3.11. Prediction of conformational B-cell epitopes

Out of 494 amino acids found within the vaccine, 245 were identified as conformational B-cell epitope residues by the Discotope 2.0 server (**Fig 6** and **S2 Table**). Depending on the number of residues, the scoring ranges of these epitopes range from 0.646 to 0.745 (**S2 Table**).

## 3.12. Molecular docking study

The molecular docking (V-TLR-2 and V-TLR-4) (V = vaccine) was executed employing the server known as Cluspro 2.0 and forty models were generated. The "V-TLR-2" complex's chosen model has the center energy score and lowest of -1277.9 KJ/mol (**Table 6**). Our "V-TLR-4" measured an energy centre score of -1028.5 KJ/mol and a lowest score of -1139.4 KJ/mol (**Fig 7** and **Table 6**). Furthermore, PyMOL and PDBsum were used to analyze and visualize the vaccine-TLR complex structures. The "V-TLR-4" complex had 298 non-bond interactions,

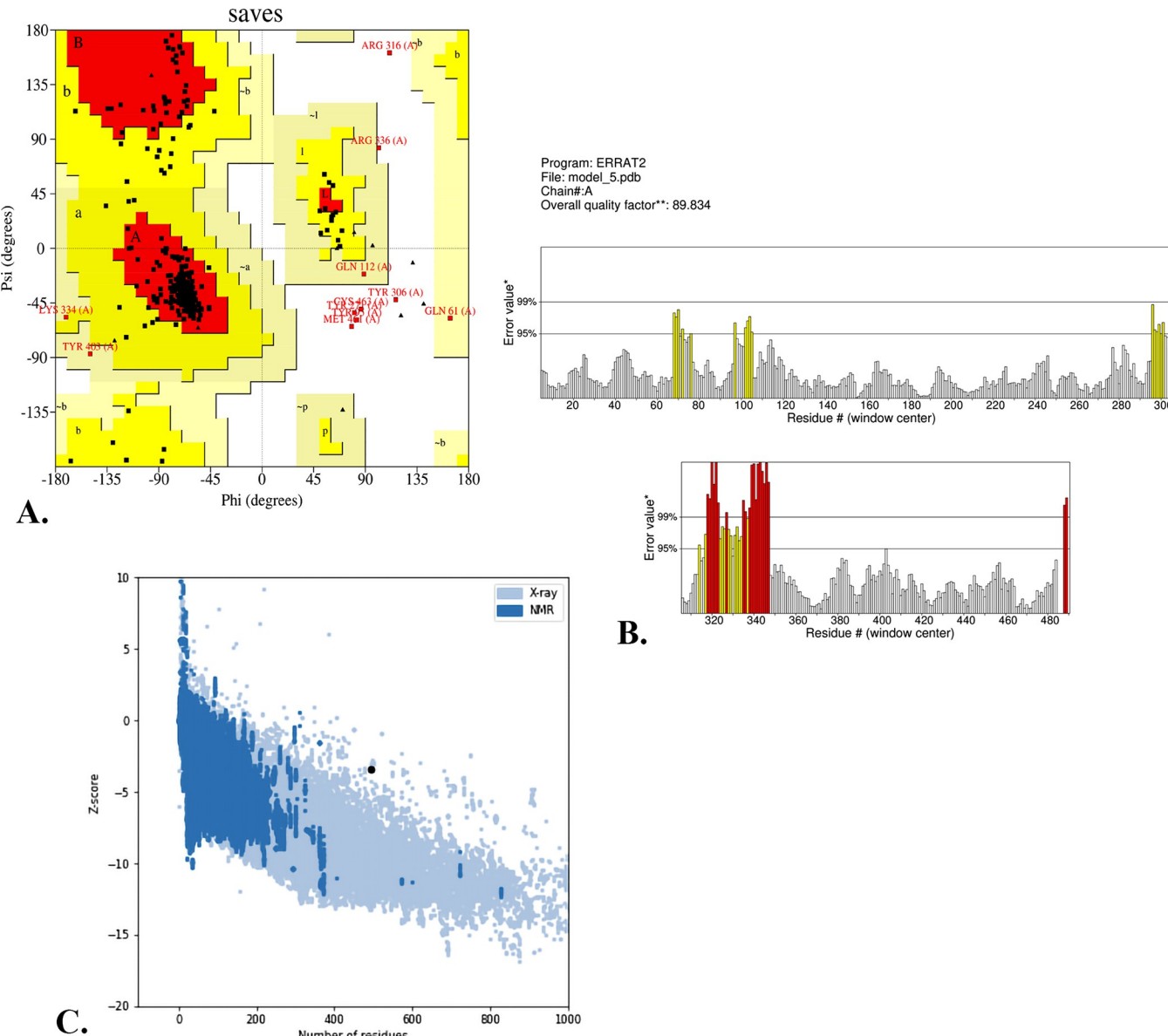

**Fig 5. The constructed vaccine was validated using I-TASSER in a model.** (A) Ramachandran plot refinement, (B) ERRAT score, and (C) Z-score.

36 hydrogen bonds, and 2 salt bridges, respectively, according to data from PDBsum. The "V-TLR-2" complex had 36 hydrogen bonds and 450 non-bond interactions (**Fig 7 and Table 6**).

### 3.13. Molecular mechanics with generalized Born and surface area solvation for free energy computation (MM-GBSA)

The HawkDock server served to determine the free binding energy (MM-GBSA) of the vaccine-receptor complexes. An estimate of the binding free energy of the "V-TLR-2" complex was generated by the server at -186.01 (kcal/mol). Consequently, the values of VDW, ELE, GB, and SA for this complex were determined to be -346.7 (kcal/mol), -860.9 (kcal/mol), 1064.38 (kcal/mol), and -42.8 (kcal/mol), respectively. In contrast, the "V-TLR-4" complex exhibited a

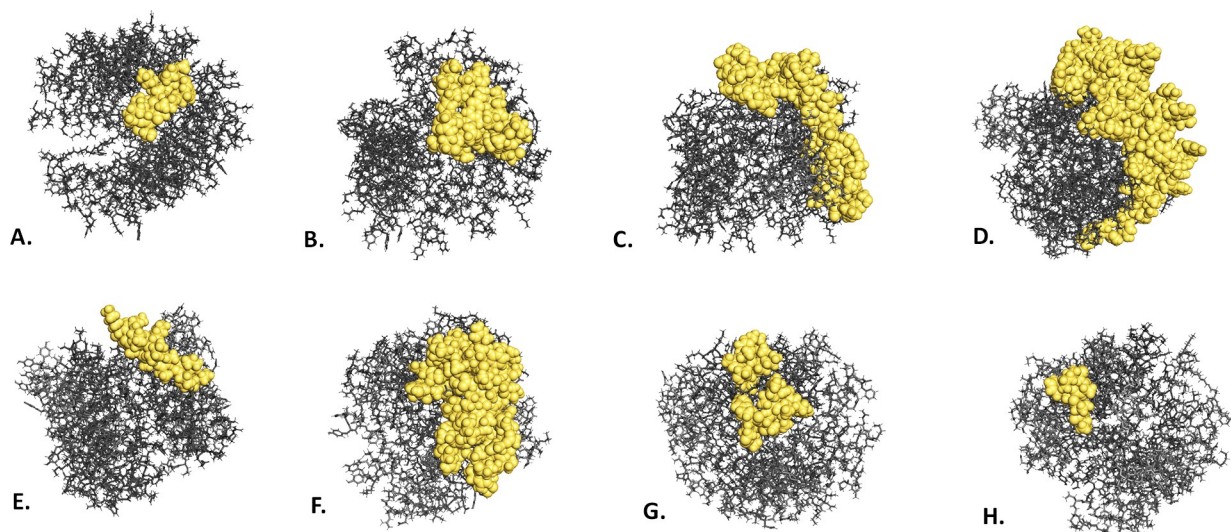

**Fig 6. 3˚ model showing the conformational B-cell epitopes of the vaccine design.** [A–H] categories. Grey sticks represent the full vaccination process, while yellow surfaces exhibit conformational B cell epitopes. Anatomical B-cell epitope modeling in three dimensions for the vaccine (A-H). Sticks in gray represent the vaccine component, while surfaces in yellow represent conformational B cell epitopes.

total binding free energy of -180.89 kcal/mol. The corresponding values for the VDW, ELE, GB, and SA were determined to be -212.27 kcal/mol, -7131.08 kcal/mol, 7190.12 kcal/mol, and -27.66 kcal/mol, respectively (**S2 Fig** and **S3 Table**).

## 3.14. Molecular dynamics simulation

**3.14.1. Root Mean Square deviation (RMSD).**    The stability of the "V-apo", "V-TLR-2", and "V-TLR-4" complexes was evaluated using the Root Mean Square Deviation (RMSD) of the backbone atoms. This plot illustrates the protein's conformational alterations in relation to its initial structure during MD simulation. The RMSD values for "V-apo", "V-TLR-2", and "V-TLR-4" were all within the range of 0.2–1 nm, 0.2–0.7 nm, and 0.2–0.7 nm, respectively, as indicated by the 3D structure of the HeV construct. After 50 nanoseconds of simulation, all structures attained stability. The RMSD plot illustrated that each of the three structures was stable (**Fig 8A, 8C and 8E**).

**3.14.2. The root-mean-square fluctuation (RMSF).**    The temporal variation of protein residues relative to a reference position is represented by the root-mean-square fluctuation (RMSF) during simulation. The HeV construct's variability was evaluated in both its free-state form ("V-apo") and bound complex configuration ("V-TLR-2", "V-TLR-4") in this study. The results suggested that no atypical fluctuations were observed in either state of the HeV candidate construct (**Fig 8B, 8D and 8F**).

**Table 6.  Docking values and associations between receptors and vaccine complex.**

| Vaccine with TLR complex | Score (KJ/mol) | | Associations between the vaccine and TLRs | | | |
|---|---|---|---|---|---|---|
| | Center | Lowest energy | Salt bridges | Disulphide bonds | Hydrogen bonds | Non-bonded contacts |
| "V-TLR-2" | -1277.9 | -1277.9 | 2 | - | 36 | 450 |
| "V-TLR-4" | -1028.5 | -1139.4 | 8 | - | 26 | 228 |

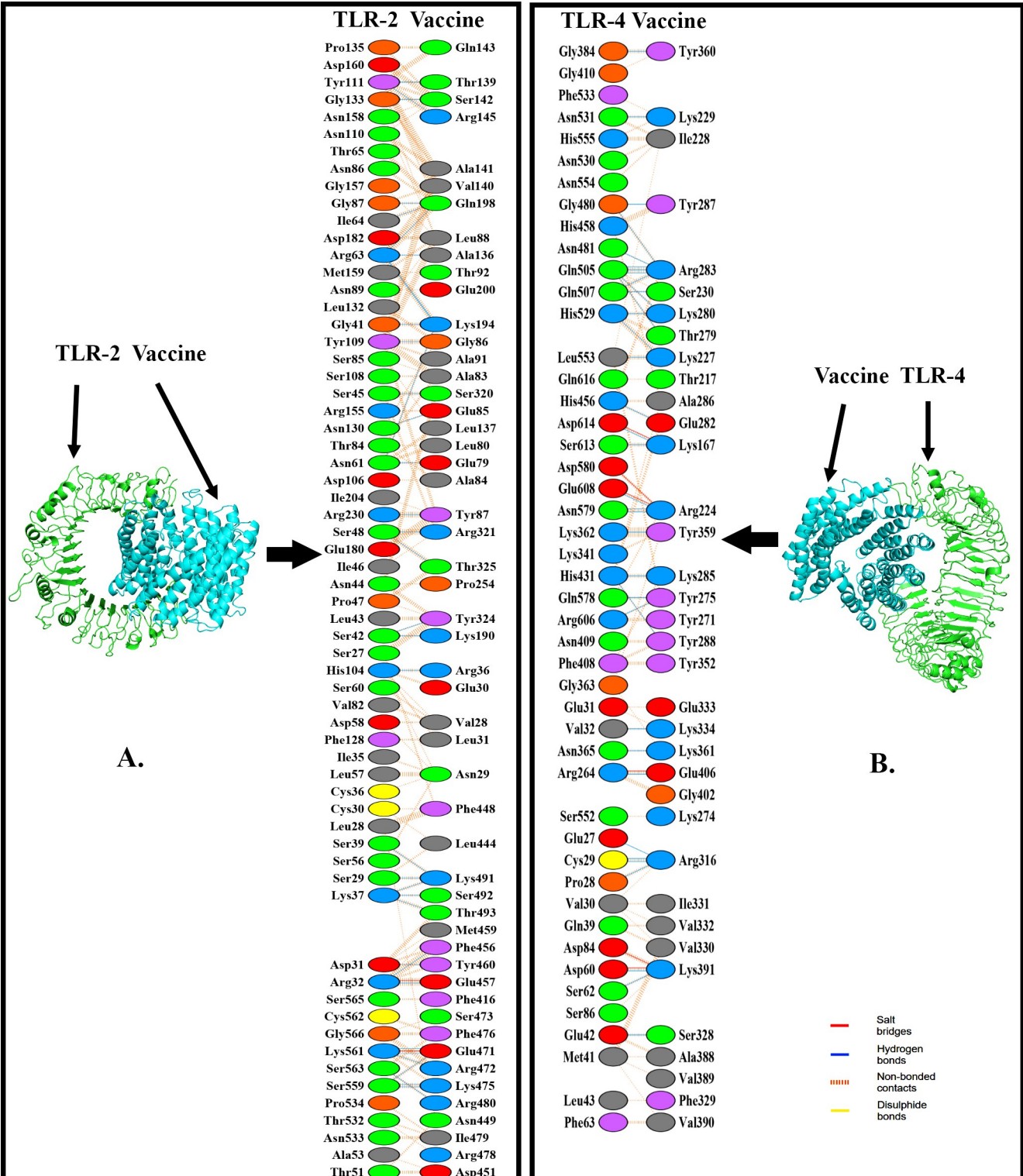

**Fig 7. The association of HeV-vaccine (cyan) and receptors (green).** In the first case, we have the vaccine-TLR-2 interaction (A) and in the second case the vaccine-TLR-4 interaction (B) is illustrated.

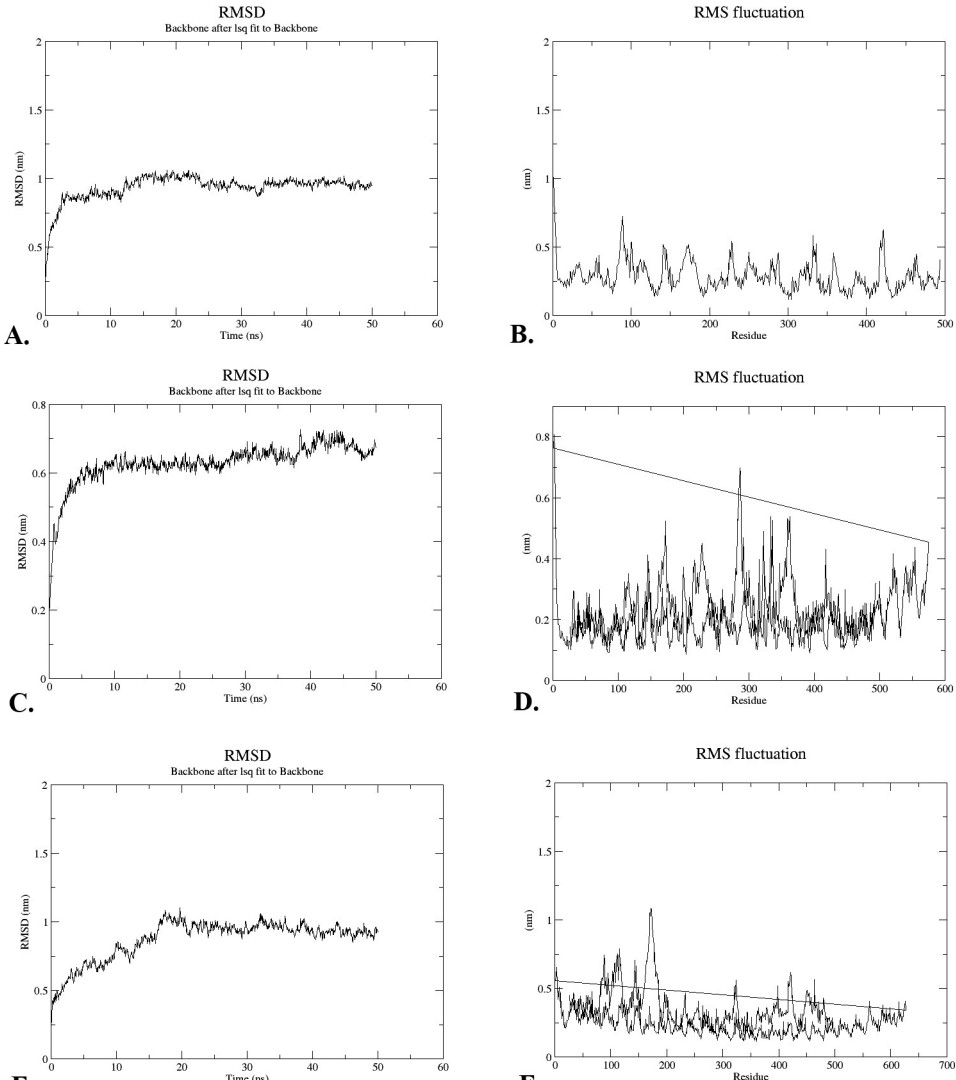

**Fig 8.** The structures from "V-apo", "V-TLR-2", and "V-TLR-4" were shown in the plots along with their computed RMSD (A, C, and E) and RMSF (B, D, and F) values.

**3.14.3. Radius of gyration.** A radius of gyration (Rg) plot was employed to evaluate the change in compactness of HeV during the simulation. Rg values were determined to be 2.55–2.8 for "V-apo" (**Fig 9A**), 3.275–3.425 for "V-TLR-2" (**Fig 9D**), and 3.325–3.46 for "V-TLR-4" (**Fig 9G**). After a 50 ns period, the Rg of both structures reached and maintained equilibrium.

**3.14.4. SASA analysis.** In addition, the SASA values projected for the structures were used to estimate the solvents within the complexes to which the hydrophobic core will be exposed. Therefore, elevated SASA values suggest that a significant portion of the protein is in contact with water, whereas diminished values suggest that a significant portion of the protein is protected within the hydrophobic core. SASA values of "V-apo" and "V-TLR-2" and "V-TLR-4" complexes were significantly different. Consequently, the V-apo structure demonstrated a mean SASA value that was substantially lower than 270 nm2 (**Fig 9B**), in contrast to the "V-TLR-2" (**Fig 9E**) and "V-TLR-4" (**Fig 9H**) structures, which exhibited SASA values

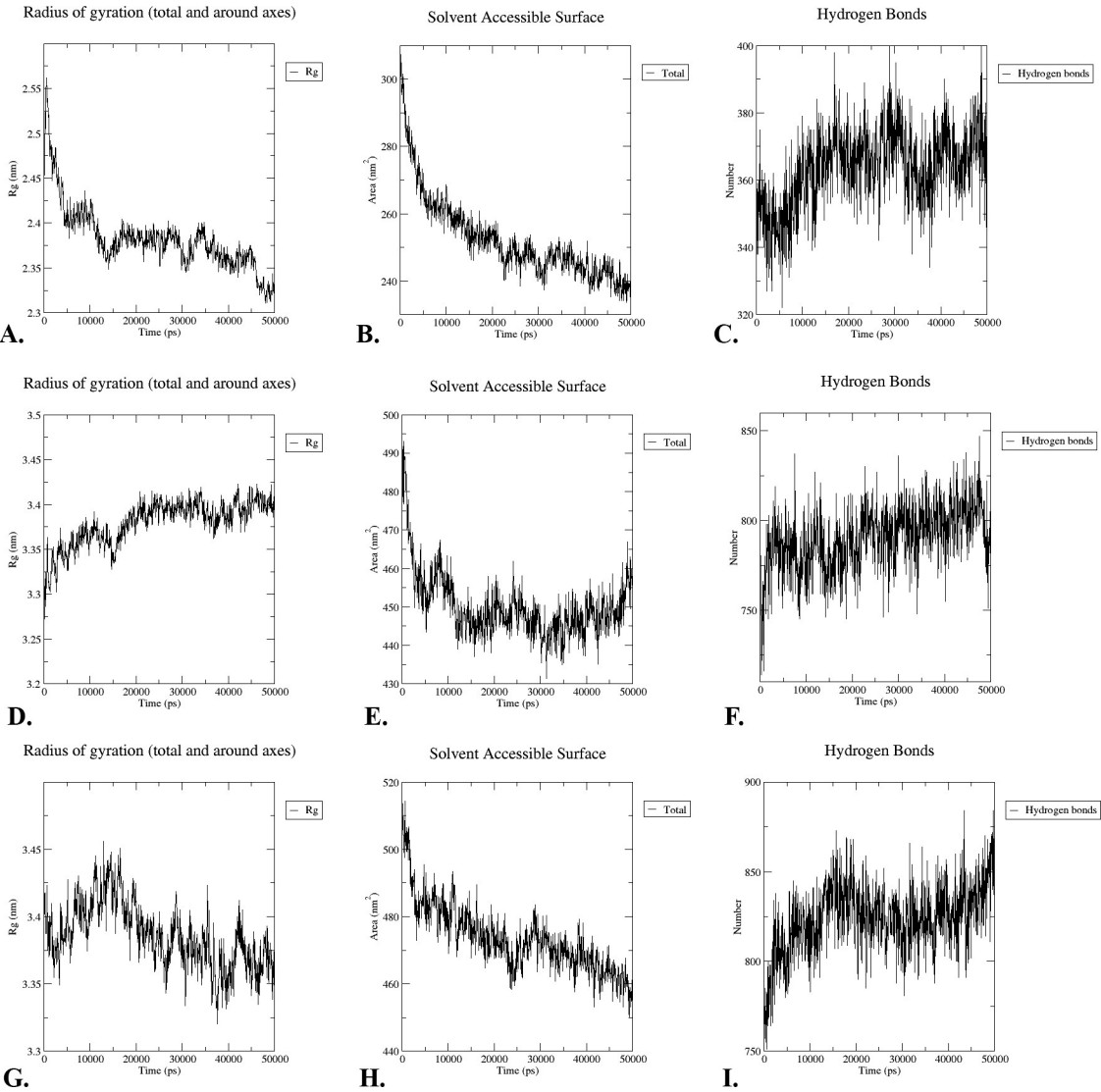

**Fig 9.** The Rg (A, D, and G), SASA (B, E, and H) and Hydrogen bonds (C, F and I) values of the "V-apo", "V-TLR-2", and "V-TLR-4" are graphically represented.

exceeding 440. However, the SASA of both complexes resumed its decline at the conclusion of the 50 ns simulation run (50000 ps).

**3.14.5. Hydrogen bonds.** In order to evaluate the "V-apo", "V-TLR-2", and "V-TLR-4" complexes, the number of hydrogen bonds between the HeV construct, TLR2, and TLR4 was determined. Furthermore, a probability distribution for hydrogen bonds was graphically represented. This plot suggests that "V-apo" contains 363.5 hydrogen bonds **(Fig 9C)**, "V-TLR-2" contains 774 hydrogen bonds **(Fig 9F)**, and "V-TLR-4" contains 818.5 hydrogen bonds **(Fig 9I)**.

## 3.15. Structure prediction of mRNA vaccine

According to the MFE score, the secondary structure of the mRNA for the vaccine was observed to have a value of -367.10 kcal/mol for the optimum structure and -250.50 kcal/mol

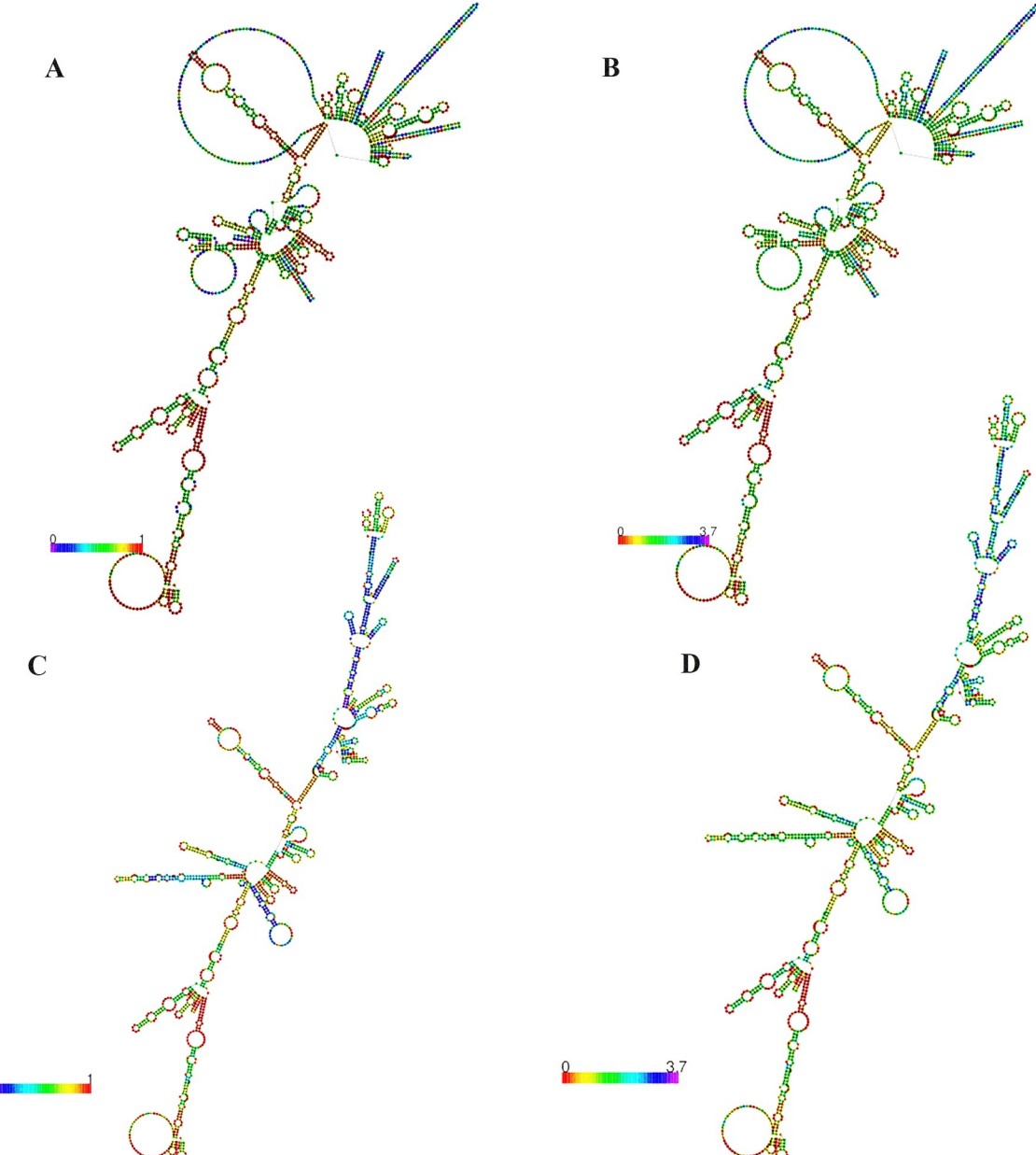

**Fig 10. Predicted mRNA structure of the vaccine RNAfold web server.** The centroid structure of the vaccine with the base pair probabilities (A), optimal structure with base pair probabilities (C), centroid structure of the vaccine with the positional entropy (B) and optimal structure of the vaccine with positional entropy (D).

for the centroid structure. In the thermodynamic ensemble, it was projected that the value would be -394.83 kcal/mol. According to, there was a correlation of 0.00% in the ensemble for the MFE structure of the vaccine **(Fig 10).** In light of this, the mRNA structure of the proposed vaccine will be stable throughout the process of entering the host, undergoing transcription, and expressing itself.

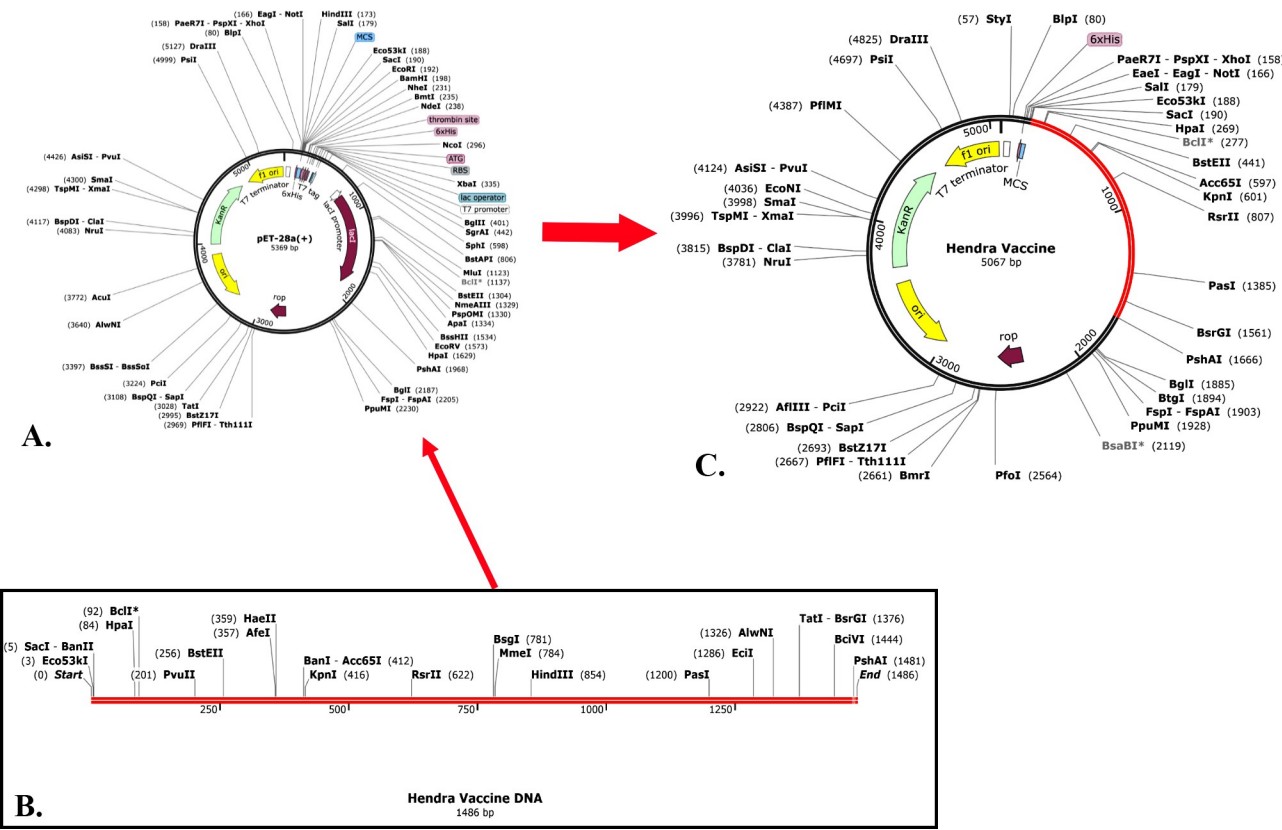

**Fig 11. Cloning the pET-28a(+) plasmid vector in a virtual environment.** Plasmid pET-28a(+) was used to insert the vaccination sequence, and the software SnapGene, which is available for free trial at https://www.snapgene.com/free-trial/, was important in this process. The Hendra Vaccine was transformed into DNA (C) using the pET-28a(+) vector (A). The red part of the cloned Hendra vaccine symbolizes the vaccine's coding genes, while the black part reflects the vector's backbone.

### 3.16. Optimization of codons and virtual cloning

The Java Codon Adaptation Tool (JCat) was implemented to optimize codons for the vaccine by increasing protein production in *E. coli* K12. A triplet sequence with 981 nucleotides was expected to have been optimised by the server. Additionally, the modified sequence's average GC content of 46.83% and codon optimization index (CAI) value of 1.0 indicate that the *E. coli* host is the most effective in promoting protein expression. Finally, the modified codon sequences were incorporated into the plasmid vector pET-28a(+) using SnapGene software, resulting in the production of a recombinant plasmid sequence (**Fig 11**).

### 3.17. Immune response simulation

A computational immunological modeling method predicted that the vaccination will produce robust adaptive immune responses. And sure enough, it did. Every single B-cell in the body showed a dramatic uptick in memory B-cell counts following three doses of vaccine. Additionally, total B-cell counts were continuously active throughout the year, which showed that immunity was maintained (**Fig 12A**). After receiving the vaccine, the body's T cell response—consisting of TH cells, TC cells, and TR cells—grew substantially. There was an increase in the concentrations of both active and memory TH cells on day 60 (**Fig 12B**). Also, active TR expression levels increased one year following the vaccination program (**Fig 12C**). Vaccination elicited robust innate immune responses in NK (**Fig 12D**), DC (**Fig 11E**), and MA (**Fig 12F**)

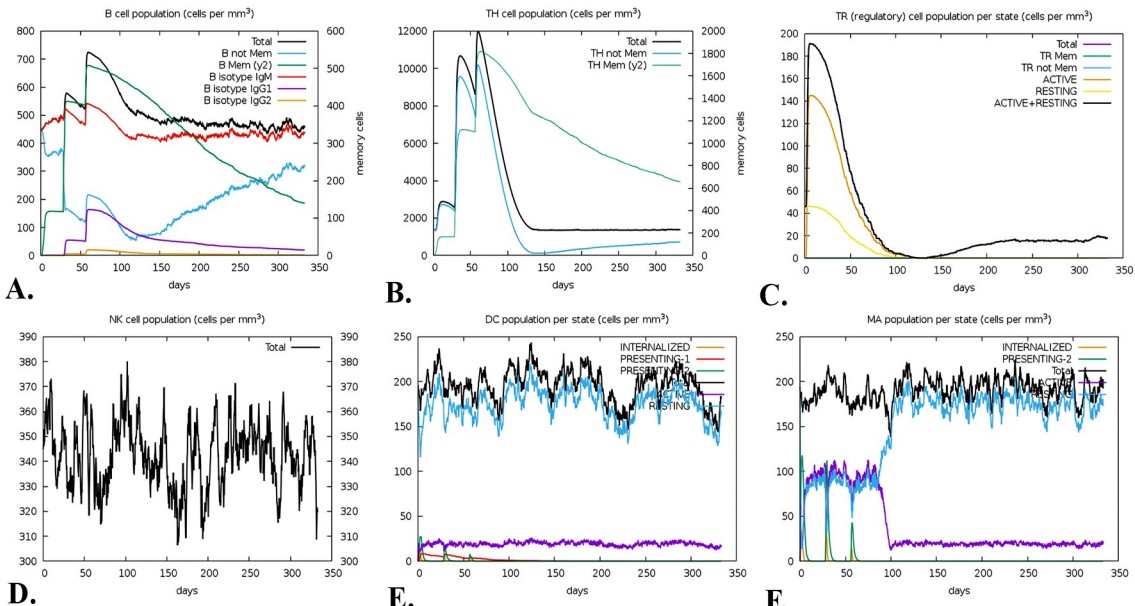

**Fig 12. A C-ImmSim-predicted vaccination immunological simulation.** In the aftermath of three successive doses, the immune response manifests as different populations of B-cells (A), TH-cells (B), TR-cells (C), NK cells (D), Dendritic cells (E), and Macrophages (F).

cells. The fact that these cells continued to function normally for an entire year implies that the immunity was unaltered. Vaccination also elicited cytokine responses that included inter-feron-g which was at highest uprising level where it was increased near to 450000 mg/ml and after 50 days it just decreased to 400000, interleukin-12 and interleukin-10 which was the low-est on this simulation only near to 50000 mg/ml, tumor growth factor-b which was 100000 mg/ml. The cytokines were detected initially, but they disappeared sixty days after the vaccine (S3C Fig). The active TC cell concentration was close to 1000 mg/ml after 50 days and it dropped sharply after 100 days (S3B and S3D Fig). The antigen count was at high peak at 550000 per cell at initial injection but it dropped significantly. The level of IgM+IgG was at the highest level at 50 days and that was about 700000 count per cell and then it dropped at 300 days (S3A Fig).

## 4. Discussion

Among the many public health initiatives, vaccination is among the most affordable, as it not only alleviates the financial burden associated with medical treatment but also extends its ben-efits to overall health, intellectual growth, and revenue generation [92]. Vaccines have played a pivotal role in mitigating infectious diseases, exemplified by the eradication of smallpox in 1980, a historic milestone [93]. They offer protection against various microbes and viruses, contributing to the well-being and development of children [94]. Furthermore, vaccines pro-mote community immunity, safeguarding vulnerable populations such as children, the elderly, and immunocompromised individuals [95]. Recent advances in sequencing and analytics have shifted vaccine development from empirical to rational design [96].

To construct a vaccine against an epidemic or pandemic causing pathogen to reduce the spread of corresponding diseases and massive death is a great challenge. Immediate vaccine preparation can play significant role during this type of situation to control the health issues. The 'in-silico' approach, which employs reverse vaccinology and immune-informatics

techniques, constitutes a single of the most effective options for this purpose. It is now more clear than ever that quick development of vaccine is necessary in response to the current danger offered by emerging infectious diseases like the COVID-19 catastrophe. The vaccination procedure has undoubtedly contributed to the enhancement of public health. Vaccination has significantly spared millions of lives, leading to a decrease in public healthcare expenses and an enhancement in the quality of life. Currently, the vaccination strategy is being implemented to combat a diverse array of infectious diseases [97,98]. Vaccines are available in a variety of forms, such as live-attenuated vaccines, killed vaccines, subunit vaccines, virus-like particles (VLPs), viral vectors, toxoids, and nucleic acid vaccines. Live-attenuated vaccines provide a high level of immunity; however, they may be detrimental to immunocompromised individuals and expectant women. Conversely, vaccines that is killed provide a more restricted level of protection and exhibit a decreased immunogenicity [99–106]. In general, subunit vaccines are safe for immunocompromised patients and can be readily modified. On the other hand, these are comparatively less immunogenic and frequently necessitate adjuvants or conjugates [107]. Even though VLPs are well-tolerated, they are subject to a complex manufacturing process and an expensive downstream process [107]. In the same vein, viral vectors demonstrate a robust immune response; however, they are susceptible to genomic integration and manufacturing process complexity [107]. Toxoid vaccines are non-virulent, stable, and do not demonstrate robust immunogenic performance [107]. DNA vaccines, a kind of nucleic acid vaccines, are considered safe and can be modified for emerging infections; nonetheless, they have the potential risk of genetic integration. mRNA vaccines represent a promising alternative to conventional vaccinations owing to their efficacy, safety, efficiency, expedited clinical development, and cost-effectiveness. They enhance antibody and cellular immunity and promptly address developing illnesses [24]. The primary component of RNA vaccines is messenger RNA (mRNA), which is intended to facilitate the translation of antigens in APCs. mRNA vaccines comprise synthetic mRNA molecules that direct the production of the antigen that will generate an immune response. *In vitro*-transcribed (IVT) mRNA mimics the structure of endogenous mRNA, with five sections, from 5′ to 3′: 5′ cap, 5′ untranslated region (UTR), an open reading frame that encodes the antigen, 3′ UTR and a poly(A) tail [108]. A potential advantage of utilizing mRNA vaccines over DNA vaccines is the potential reduction in the risk of adverse effects and autoimmune disorders due to the rapid degradation of messenger RNA (mRNA) vaccines. Another benefit of mRNA vaccines is that they are incapable of causing cancer due to their lack of incorporation into host DNA. Additionally, they can be produced rapidly and easily, and they activate the body's natural defense mechanisms, which create an impenetrable barrier against infections by producing antibodies and cellular immunity [23,24]. The initial mRNA vaccine for infectious diseases was produced in 2012 to combat influenza A. In mice, *in-vivo* experiments demonstrated that specific B- and T-cells based protection was formed. Ferrets and piglets were also used to demonstrate the vaccine's effectiveness [109]. Subsequently, different mRNA vaccines were evaluated in animal models for their efficacy against the Zika virus, Ebola virus, cytomegalovirus, and human immunodeficiency virus (HIV), among other viruses [110–115]. HIV currently impacts 38 million individuals worldwide and is expected to affect as many as 42 million individuals by 2030. Isolating broadly neutralizing mAbs from infected individuals who neutralize multiple HIV genotypes is a novel vaccination strategy against HIV. The VRC01 broadly neutralizing mAbs have recently garnered attention for their capacity to neutralize 98% of HIV strains and prevent the transmission of antibody-sensitive strains with 75.4% efficacy [116,117]. This is particularly noteworthy. In a single study, a 0.7 mg kg−1 intravenous injection of a nucleoside-modified mRNA encapsulated in LNP that expressed VRC01, produced antibody concentrations that were comparable to those typically attained by injecting a 10–20 mg kg−1 dose of mAb

protein. Significantly, rodents were safeguarded from intravenous HIV-1 challenge by a single dose [118].

The FDA recently issued authorization for the first two SARS-CoV-2 mRNA vaccines, specifically BNT162b2 from Pfizer/BioNTech and mRNA-1273 (Spikevax) from Moderna. Additionally, the FDA has granted commercial approval to BNT162b2 by Pfizer/BioNTech, which is the first mRNA vaccine. Professors Katalin Kariko and Drew Weissman were awarded a "Nobel prize" in physiology or medicine in 2023 for their contributions to the development of SARS-CoV-2 mRNA vaccines. During the COVID-19 pandemic, millions of doses of these mRNA vaccines have been administered globally. This has facilitated the compilation of the efficacy and safety data on these vaccines [119]. The effectiveness advantages of the BNT162b2 and mRNA-1273 vaccines were immediately evident upon their initial delivery to the general population. The administration of these vaccinations was promptly linked to a reduction in COVID-19 symptoms and transmission [120]. Moderna is currently conducting an assessment of three single-dose vaccine candidates that encode the prefusion F protein of Respiratory Syncytial Virus (RSV): mRNA-1172 and mRNA-1777 for adults, and mRNA-1345 for children. In phase I clinical trials, mRNA-1777 elicited a robust humoral response with RSV neutralizing antibodies, a CD4+ T cell response to RSV F peptides, and no severe adverse events. mRNA-1345's sequence has been further engineered and codon-optimized to improve its immunogenicity and translation in comparison to mRNA-1777 [108]. Interim phase I data indicate that a 100 μg dose of mRNA-1345 generates neutralizing antibody titers that are approximately eightfold higher than those of mRNA-1777 one month following vaccination. Moderna's ultimate objective is to combine mRNA-1345 with its pediatric human metapneumovirus/parainfluenza virus type 3 (hMPV/PIV3) candidate mRNA-1653 in order to immunize children against three distinct pathogens with a single formulation [108,121]. The emergence of these innovative vaccines has marked a new era of innovation in the field of vaccination against cancer and infectious diseases [122–124]. Furthermore, these techniques have been effectively utilized in the invention of vaccines against a several kinds of infectious diseases, such as Zika, chikungunya, and influenza [125]. Reverse vaccinology has been successfully used to extract surface-associated proteins from pathogen genomes without culturing the pathogens, and it is a trustworthy and secure approach [125]. These success stories implement the motivation to design vaccines or other therapies to design against different pathogens and cancers by utilizing bioinformatics. Hopefully, in the next five years, the use of bioinformatics will be more enhanced [126,127]. By June 2023, three vaccine candidates (HeV-sG-V, PHV02, and mRNA-1215) and one mAb (m102.4) had completed a registered human clinical trial. The trials were phase 1, dose-ranging studies that were conducted in the United States of America or Australia and enrolled healthy adults. The Target Product Profile's dose regimen and route of administration criteria are met by all vaccine candidates. However, future evaluations of other criteria, including efficacy and reactogenicity, will be necessary as new evidence becomes available [128]. Nonetheless, there is still a significant void in vaccine coverage for Hendra virus (HeV), even though reverse vaccinology has been successful in developing vaccines. At present, there are no mRNA vaccines that have been approved for the Hendra virus. We were greatly inspired to develop a novel mRNA vaccine against the Hendra virus due to the efficacy and safety of mRNA vaccines, which have been demonstrated by the numerous successful stories and the development of the BNT162b2 and mRNA-1273 vaccines.

The current study employed additional methods to enhance vaccine prediction accuracy. An mRNA vaccine targeting the G (Glycoprotein), F (Fusion protein), and M (Matrix protein) proteins, known to be potent antigenic components of the HeV virus, was successfully developed. Epitopes originating from the G, F, and M proteins were predicted and included in the vaccine formulation. We tested the antigenicity, allergenicity, and toxicity of each CTL epitope

that was chosen. Selected linear BCL epitopes showed little to no allergenic properties, although immunogenicity and the capacity to stimulate IFN-γ production were validated for HTL epitopes. The ultimate multiepitope vaccine was generated by combining several types of adjuvants and linkers that targeted specific epitopes. In all, the vaccine reached 100% of the world's population, with individual coverage rates of 91.81% and 98.55%. Several regions were covered extensively, including the Americas, Eurasia, Oceania, Southeast Asia, Central Africa, and East Africa.

The vaccine candidate is considered suitable for vaccination due to its molecular weight of 55349.73 Da and its stability in biological environments, as evidenced by its 37.18 instability index. Molecules with an instability index below 40 are considered stable [129]. The vaccine was also regarded as significantly soluble. The solubility was also to evaluated in overexpressed *E. coli.* which was essential for biochemical and functional investigations [90,130]. The GRAVY score was found to be -0.503, suggesting a mildly hydrophilic nature. The theoretical pI of the vaccine is 9.76, which implies that it possesses acidic properties. It also has aliphatic index of 82.11. This value suggests that the vaccine demonstrated hydrophobic characteristics that were consistent with the reported inclusion of aliphatic side chains and the vaccine is thermostable [131]. Developing an efficacious vaccine necessitates the consideration of the protein folding process into its 2˚ and 3˚ structures. For protein-specific immune responses, antigens contained in unfolded proteins and α-helical coils play a crucial role. Hence, antibodies readily develop in response to unforeseen infections, specifically targeting these two structural antigens [132]. After that, we were able to accurately forecast the vaccine's 2˚ and 3˚ structures, which led to something positive outcome. The vaccine's 2˚ structure is quite stable since it contains many alpha helices, beta sheets, and coils. Also, the 3˚ structure of the vaccine was guaranteed because a large fraction (88.5%) of the residues were found in the most favorable locations, as shown by the Ramachandran diagram analysis. Calculated to be -3.44 KJ/mol, the Z-score further indicates the vaccine model's overall quality as positive Z-scores typically indicate inaccurate or problematic components of a model [133]. Five mutations were found at residue pairings of SER 5-ASP 8, GLU 117-ALA 120, LYS 246-METT 265, ILE 311-LYS335, and LEU 412-GLY 432, after the other parameters were reduced into two pairs using the Disulfide by Design website. The probability of a vaccine-immune cell binding relationship was determined through a docking investigation that employed TLR-2 and TLR-4. Based on these findings, it appears that the vaccine binds strongly to the Toll-like Receptor-2 (TLR-2) (with a lowest energy score of -1277.9 KJ/mol) and Toll-like Receptor-4 (TLR-4) (with a lowest energy score of -1139.4 KJ/mol) receptors. The "V-apo", "V-TLR-2", and "V-TLR-4" showed remarkable stability in the molecular dynamic simulation, which revealed a wide range of RMSD, RMSF, Rg, SASA, and hydrogen bonds. The equilibration and structural stability of the protease in the presence of a bound ligand are examined using the RMSD parameter, which is the rate of the mean distance between the atoms. A lower RMSD value of the ligand-docked protein than the solitary protein demonstrates the structural stabilization of the complex [134,135]. At 50 ns the RMSD values were lower than 1 for "V-apo", "V-TLR-2" and "V-TLR-4" respectively. This suggests that the all the complexes were stable. The dynamics of a complex with a low RMSF may be more constrained, while a complex with a high RMSF is anticipated to be more flexible. The RMSF is a metric that quantifies the structural displacement of an amino acid from its average position during the simulation. The RMSF method is effective in evaluating the local flexibility of protein structures and distinguishing between flexible and rigid regions. The result demonstrated that the RMSF values were also lower than 1 approximately for "V-apo", "V-TLR-2" and "V-TLR-4" respectively at 50 ns. This result suggests that all the complexes were stable. The solvent-accessible surface area (SASA) of proteins has long been regarded as a key component in investigations of protein stability and folding. Around a

protein, this surface is defined as the one that meets the van der Waals contact surface and is bounded by a solvent sphere with a hypothetical center. Additionally, a higher SASA value suggests that a greater proportion of the protein's surface area is exposed to the solvent, potentially resulting in an increase in the protein's solubility [136]. The SASA values for "V-apo", "V-TLR-2", "V-TLR-4" were 240 nm$^2$, 455 nm$^2$ and 460 nm$^2$ respectively at 50,000 ps, suggesting that "V-TLR4" was accessible to their solvent environments at the highest level. The compactness of the protein structure is determined by the radius of gyration (Rg). The tightest packing characteristic of proteins, which ensures stability, is determined by the lowest radius of gyration. The lower the Rg value, the more electrostatic and hydrophobic contacts are possible between compounds [137]. In the radius of gyration analyses of "V-apo", "V-TLR-2" and "V-TLR-4", the Rg values were high for all the complexes at 0 ps but as time passed the values followed a downward trend and the values 2.325, 3.4 and 3.375 respectively. The order of stability is arranged as: "V-apo" > "V-TLR-4" > "V-TLR-2". The formation and stabilization of protein structures are significantly influenced by hydrogen bonds (H-bonds) [138]. The H-bond analysis revealed that "V-apo", "V-TLR-2" and "V-TLR-4" had 370, 800 and 875 H-bonds at 50000 ps suggesting the order of stability as follows: "V-TLR-4" > "VTLR-2" > "V-apo". Overall, the molecular dynamic simulation experiments demonstrated that all three complexes functioned satisfactorily. The minimum free energy of the mRNA vaccine was predicted to be -367.10 kcal/mol (optimum structure) and -250.50 kcal/mol (centroid structure) indicating the stability of the vaccine following its entry, transcription, and expression in the host. The purpose of employing codon optimization was to examine the recombinant vaccine's expression in the *E. coli* strain K12. With a GC content of 46.83% and a CAI score of 1.0, the test findings indicate that the vaccine showed a considerable amount of expression in the vector. The vaccine can stimulate simultaneously the innate and adaptive immune responses, according to empirical study. It was clear that memory B-cells and T-cells existed, and the immunity that the B-cells maintained for a year was also noticeable. The noticeable features of TH cell activation and the release of IFN-γ and IL-2 were demonstrated in the first expansion of concentrations of these factors after the first injection, which kept rising with each subsequent exposure to the antigen. In addition, the data that was gathered revealed the production of immunoglobulins (IgM and IgG), which function as markers for a humoral immune response. However, a number of HeV vaccines have recently been designed targeting the specific epitopes or peptide antigens from the proteome of the virus. Ahmad F et al. 2022 designed an antiviral drug targeting the G protein of HeV [139]. In contrast to the mentioned drug, the outcomes of our developed vaccines were thought to be satisfactory because firstly we designed an mRNA vaccine which is superior to their subunit vaccine. Next, we used both F (fusion) and M (matrix) proteins along with G (attachment glycoprotein) protein which may give stronger immune response after administrating into body. As we designed vaccine, we conducted population coverage study, physiochemical study, 2° structure prediction, prediction, improvement, and validation of 3° structures, disulfide engineering, conformational B-cell epitope prediction, *in-silico* cloning and immune simulation. We also used human TLR-2 & TLR-4 for molecular docking to enhance vaccine's potency. In all these steps, our vaccine candidate provided promising results and that is why our tailored therapy is better than their study. Hossain, Afrin Sultana *et al.* 2021 designed a multiepitope based vaccine against HeV utilizing seven CTL, five HTL, and five linear BCL epitopes from the highest antigenic proteins, namely glycoprotein, fusion protein, and nucleoprotein. Stability, significant binding affinity for the TLR-4 receptor, and population coverage of 92.8% for CTL and 34.78% for HTL epitopes, respectively, were demonstrated in the suggested vaccination using docking and dynamics simulation studies [5]. While docking with TLR-2 and TLR-4 receptors, we used eighteen T alongside B cell epitopes of the proteins named G, F, and M; our population coverage for CTL

epitopes was 98.55% and for HTL epitopes it was 81.81%. Our vaccine is an mRNA-based vaccine with potential result in structures, physicochemical properties, docking, immune simulation and molecular dynamic simulation which make it superior to subunit vaccines. Because of this, our research can be seen of as an improvement over their research. Kamthania M. et al. 2019 designed a HeV vaccine by targeting twenty two nucleocapsid, twenty matrix, twenty two fusion, twenty two glycoprotein, fourteen W protein, seventeen V protein, fifteen C protein and fifteen polymerase protein of HeV and they did epitope conservancy test and also conducted dynamics simulation in 10 ps [140]. On the other hand, we targeted three vital epitopes of HeV (G, M and F) and designed an mRNA-vaccine based on CTL, HTL and linear BCL epitopes and also conducted the dynamics simulation at 50000 ps where it resulted in that our vaccine was stable. Because of this, our study is considered more credible than theirs.

The capability of the produced HeV multiepitope mRNA vaccine to trigger strong immune responses from both cells and humor through G, M, and F protein epitopes is the primary emphasis of this study. This study lacks credibility because it has not been validated experimentally through *in-vitro*, *in-vivo*, or clinical trials.

## 5. Conclusion

Messenger RNA (mRNA) vaccines have evolved from a concept that elicited criticism to a clinical reality over the past several decades. The most rapid vaccine development in history was catalyzed by the COVID-19 pandemic in 2020, with mRNA vaccines at the vanguard of efforts. Although it is now evident that mRNA vaccines can provide patients with rapid and secure protection from infectious diseases. Developing an mRNA vaccine against Hendra virus (HeV) using next-generation reverse vaccinology technologies has significant potential in combating this deadly pathogen. By employing sophisticated computational algorithms in conjunction with the methodical identification of antigenic targets and epitopes, this novel strategy may effectively stimulate robust immune responses while minimising detrimental consequences. When compared to conventional approaches, the rational design of a vaccine candidate provides benefits in terms of safety, efficacy, and scalability. The results showed that the vaccine was constructed only with antigenic, non-allergic and non-toxic epitopes and the chosen epitopes exhibited a high level of population coverage in a variety of regions around the world. The vaccine was acidic, hydrophilic and soluble. Subsequently, the docking analysis demonstrated that the vaccine has a high binding affinity for TLR-2 and TLR-4 receptors, and the MM-GBSA analysis corroborated this assertion. The vaccine construct was subjected to molecular dynamics simulation analyses, which revealed a high level of binding affinity and stability between the construct and the receptor. This finding may indicate that the *in-vivo* interaction and immune responses will be positive. In addition, the vaccine was tightly expressed in a bacterial vector that was computationally designed. In terms of immunological responses, the vaccine exhibited both humoral and adaptive immunity. Ultimately, this mRNA vaccine would remain stable enough following its entry, transcription, and expression in the host. All of the exhaustive results suggest that the mRNA vaccine that has been constructed is a promising and efficient candidate. Consequently, *in-vitro* investigations, animal model testing, and human testing are required to guarantee the safety as well as effectiveness of this vaccine candidate. Such research would culminate in the creation of a powerful instrument for the prevention and management of Hendra virus infections in both human and animal populations.

## Supporting information

**S1 Fig. The 3D disulfide engineering of the structure of the multi-epitope vaccine.** (A) The wild type; (B) The five introduced disulfide bonds are represented by yellow sticks to denote

the mutant form.
(TIF)

**S2 Fig.** The "V-TLR-2" (A) and "V-TLR-4" (B) complexes were analyzed using MM-GBSA. The cyan ribbon represents the vaccine, and the green ribbon the receptors.
(TIF)

**S3 Fig. A C-ImmSim-predicted vaccination immunological simulation.** In the aftermath of three successive doses, the immune response manifests as different populations of B-cells (A), Antigen (B), TC-cells per state (C), Cytokines (D), TC cells.
(TIF)

**S1 Table. Pairs of vaccine residues that can form disulfide bonds are listed here.**
(DOCX)

**S2 Table. Discotope 2.0 predicted the conformational B-cell epitopes residues of the vaccine structure.**
(DOCX)

**S3 Table. The MM-GBSA analysis of the "V-TLR-2" and "V-TLR-4" complexes.**
(DOCX)

## Author Contributions

**Conceptualization:** Ahmad Abdullah Mahdeen, Imam Hossain.

**Data curation:** Ahmad Abdullah Mahdeen, Imam Hossain, Md. Habib Ullah Masum.

**Formal analysis:** Ahmad Abdullah Mahdeen, Imam Hossain.

**Investigation:** Imam Hossain, Md. Habib Ullah Masum.

**Methodology:** Ahmad Abdullah Mahdeen, Imam Hossain, Sajedul Islam.

**Software:** Ahmad Abdullah Mahdeen.

**Supervision:** Imam Hossain.

**Validation:** Ahmad Abdullah Mahdeen, Md. Habib Ullah Masum.

**Visualization:** Md. Habib Ullah Masum.

**Writing – original draft:** Ahmad Abdullah Mahdeen, Imam Hossain, Sajedul Islam, T. M. Fazla Rabbi.

**Writing – review & editing:** Ahmad Abdullah Mahdeen, Imam Hossain, T. M. Fazla Rabbi.

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
