## [Decision Letter · Decision Letter 0]

22 Jul 2024

PONE-D-24-26158Designing a Chimeric Multiepitope Vaccine Against Hendra Virus (HeV) Using Immunoinformatics, Reverse Vaccinology, and Molecular Dynamics SimulationPLOS ONE

Dear Dr. Hossain,

Thank you for submitting your manuscript to PLOS ONE. After careful consideration, we feel that it has merit but does not fully meet PLOS ONE’s publication criteria as it currently stands. Therefore, we invite you to submit a revised version of the manuscript that addresses the points raised during the review process. Please submit your revised manuscript by Sep 05 2024 11:59PM. If you will need more time than this to complete your revisions, please reply to this message or contact the journal office at plosone@plos.org. Please include the following items when submitting your revised manuscript:A rebuttal letter that responds to each point raised by the academic editor and reviewer(s). You should upload this letter as a separate file labeled 'Response to Reviewers'.A marked-up copy of your manuscript that highlights changes made to the original version. You should upload this as a separate file labeled 'Revised Manuscript with Track Changes'.An unmarked version of your revised paper without tracked changes. You should upload this as a separate file labeled 'Manuscript'.

We look forward to receiving your revised manuscript.

Kind regards,

Rajesh Kumar Pathak, Ph.D.

Academic Editor

PLOS ONE

Journal Requirements:

2. Please note that Table 6 is included in the manuscript, while Table 5 is missing. 

**Additional Editor Comments:**

Several shortcomings were identified during the review process that raise concerns about its suitability. Additionally, the manuscript does not sufficiently address possible research gaps. The quality of the figures, particularly those related to molecular dynamics simulation analysis, requires extensive revision. The RMSF plot, in particular, needs corrections.

Reviewers' comments:

Reviewer's Responses to Questions

**Comments to the Author**

1. Is the manuscript technically sound, and do the data support the conclusions?

Reviewer #1: Yes

Reviewer #2: Partly

Reviewer #3: Yes

2. Has the statistical analysis been performed appropriately and rigorously? 

Reviewer #1: Yes

Reviewer #2: Yes

Reviewer #3: Yes

3. Have the authors made all data underlying the findings in their manuscript fully available?

Reviewer #1: Yes

Reviewer #2: Yes

Reviewer #3: Yes

4. Is the manuscript presented in an intelligible fashion and written in standard English?

Reviewer #1: No

Reviewer #2: No

Reviewer #3: Yes

5. Review Comments to the Author

Reviewer #1: 1. Author used the term ‘V-apo’ in many places of this manuscript. What does ‘V-apo’ denotes?

2. “Codon optimization for Escherichia. coli synthesis allowed…”- here the part ‘Escherichia. coli’ should be written as ‘Escherichia coli’. Their should be no full stop between Escherichia and coli

3. “Among the Paramyxoviridae family of viruses is the genus Henipavirus”- in the sentence the family and genus should be italicized.

4. “The P gene's co transcriptional mRNA editing produces V and W proteins…”- what does the P, V and W indicate?

5. “… attract viral glycoproteins, viral RNPs…”- what does the RNPs mean?

6. In the manuscript, you can use the term ‘linear BCL’ instead of ‘LBL’, as the ‘LBL’ also represents the term ‘lymphoblastic lymphoma’.

7. “Reverse vaccinology involves identifying antigenic…”- correct the grammar (“involves in identifying”).

8. Italicized the words: in-silico, in-vitro, in-vivo.

9. On the basis of which parameters, these proteins were selected, like- homologous with human & mice proteins, antigenicity, allergenicity, toxicity, etc. Explain.

10. “The full amino acid sequences of the Henipavirus hendraense (HeV)…”- italicized the part ‘Henipavirus hendraense’.

11. In the CTL epitopes prediction, mention the parameters with input values, i.e. set here, like- peptide length, selected alleles, threshold for strong binders: % Rank, threshold for weak binders: % Rank, etc. And why not you use the other well-known alternative- NetCTL 1.2? Is there any specific reason or advantages using NetMHC-4.0 and IEDB server? How did you combine the results from both servers in this study?

12. What threshold value was set to compute the antigenicity in VaxiJen 2.0 server?

13. Mention the input parameters i.e. used to predict the HTL and linear BCL epitopes.

14. “… their respective databases to learn about the allergenicity, toxicity, and antigenic properties of the selected epitopes.”- the ‘allergenicity’, ‘toxicity’ and ‘antigenic properties’ are written in wrong sequence. Please check again and correct it.

15. Mention the selected areas or regions and alleles used in population coverage analysis.

16. Why did you use the such linkers ‘EAAK, AYY, AK, KFER’ to join the epitopes. What are the significances of them? Please explain.

17. “Two more sites (http://scratch.proteomics.ics.uci.edu (49) and http://www.ddg-pharmfac.net/vaxijen/VaxiJen/VaxiJen.html) (37), were used to verify the antigenicity of the vaccine construct”- write the proper names of the web-servers before the URLs.

18. “We identified potential cysteine disulfide bonding pairs within the vaccine structure using the Disulfide by Design 2.12 algorithm…”- Check the version. It may be ‘2.13’.

19. Mention the input parameters used in ElliPro server.

20. In the Cluspro 2.0 server, information should be provided which chain authors have used: single chain or both chains in case of TLRs?

21. Why did you use PDBsum with PyMOL? Is there any necessity?

22. “We used the MM-GBSA methods, which rely on molecular mechanics and the Generalized Born method…”- what does this ‘Generalized Born method’ mean?

23. “This server utilizing the Escherichia. Coli: K12 codon framework can assess the protein's expression level by…”- write ‘Escherichia. Coli’ as ‘Escherichia coli’.

24. “For the purpose to clone the optimized vaccine gene sequence into the E. coli plasmid vector pET-28a(+)”- italicize the part ‘E. coli’ as ‘E. coli’.

25. Why did you select this pET-28a(+) vector? Is there any significance to choose these restriction sites (Ecp53kI and PshAI)?

26. “… a regimen consisting of three doses was established…”- why three doses were required? What was the time duration during this study for immune simulation? In which time points the mentioned three doses were introduced?

27. “… and the duration of the steps (one, eighty-four, and one hundred eighty-six), were initially assigned to their respective default values.”- how did author calculate these values?

28. The images’ quality is/are very poor in resolution. Author should provide high resolution images.

29. In the table 1, mention the start and end points of the CTL peptides, also mention the alleles used here.

30. In the table 2, mention the start and end points of the HTL peptides.

31. In the table 3, mention the end points of the BCL peptides.

32. In the population coverage analysis, why did author selected only specific countries, and why not world population instead of these countries? And what are the alleles considered here?

33. What was the basis for selecting the adjuvant ‘50S ribosomal protein L7/L12’?

34. “The SOLpro server suggested that the vaccine had the potential to dissolve effectively when expressed in E. coli, as evidenced by its score of 0859720.”- recheck the value, it may be ‘0.859720’.

35. “The SOPMA server predicted that the vaccine's 2º structure would 369 consist of a random coil (23.68%), extended strands (18.22%), and an alpha helix (53.64%) (S1 Table, Figs 4B and 4C).”- re-write the S1 Table as ‘Table 5’ and change accordingly (as you put the same table in supporting documents also).

36. “Conversely, the preliminary model's score of -2.69 undoubtedly suggests that the projected model is more accurate (Fig 5C).”- explain in detail, in what basis you had concluded that the latest refined model was better.

37. In the immune simulation, some graphs are missing, i.e. Antigen and immunoglobulins, CD8 T-cytotoxic lymphocytes, Cytokines. These graphs are important to describe the immune responses, please add these in the manuscript and discuss their results.

38. “Vaccines against this virus were developed using RV and …”- use the full form of ‘RV’ at least once in the manuscript, like- ‘reverse vaccinology (RV)’.

39. “… as evidenced by its aliphatic index of 8211…”- re-check the value, it may be ‘82.11’.

40. “Calculated to be -3.44 KJ/mol, the Z-score further indicates the vaccine model's overall quality.”- how does the Z-score indicate the overall quality? Explain.

41. “Ahmad, F, Albutti A, Tariq, et al. 2022 designed an antiviral drug targeting the G protein of HeV (93).”- re-write this as “Ahmad F et al. 2022 designed…”.

“Kamthania, M., Srivastava, S., Desai, M. et al. 2019 designed a HeV vaccine …” as “Kamthania M. et al. 2019 designed …”.

42. “Because it is a preventative treatment that encourages the immune system to work naturally against HeV, our tailored therapy is better than their study.”- Kindly explain in what way you had concluded this statement in favour of your study.

Reviewer #2: Referee: 1

Comments to the Author

Considering impact of viruses on public health, information could play a significant role if there is any novelty in the work. Manuscript needs major revision before publication.

Comment 1: Introduction section is not appealing and a substantial analysis with validated approaches are crucial. For example, what question you want to answer and what research has been done in last two years on this. Discussion section is very generic and needs to be improved with logical reasoning. How do you see this area unfolding in the next 5 years?

Comment 2: There are novel variants of Hendra virus, kindly explain the significance (if any) of your vaccine in combating infection with the variants also.

Comment 3: The author has used Hawk Dock server to determine the free binding energy (MM-GBSA) of the vaccine-receptor complexes. Validation of this energy with some other simulation softwares such as NAMD should be performed.

Comment 4: Author has used I-TASSER to generate the tertiary structure and not AlphaFold-cohab. AlphaFold2 has facilitated the rise of structure prediction performance to new heights, regularly competitive with experimental structures in CASP14. Therefore, the reason for not using alphafold2 or RaptorX should be mentioned in the methods section.

Comment 5: Include the limitation of your study.

Comment 6: Add new references (last 2 years) regarding the use of bioinformatics and generation of subunit vaccine. Cite these articles PMID: 38117103, PMID: PMID: 38782944 along with other recent articles). If any vaccine has gone from bioinformatics to clinical side, write about that to show significance of bioinformatics as well as your work in generating vaccine in the real life scenario.

Reviewer #3: Manuscript explained workflow for developing multiepitope Hendra virus vaccine using already standardized protocol available in literature.

Already single dose investigational subunit vaccine for human use against Nipah virus and Hendra virus is available as per current literature. Authors have to clearly mention how their designed multiepitope vaccine is superior to the subunit vaccine already developed.

More information on parameter selection and plot interpretation is required for immune response simulation.

6. PLOS authors have the option to publish the peer review history of their article (what does this mean?). If published, this will include your full peer review and any attached files.

Reviewer #1: No

Reviewer #2: No

Reviewer #3: **Yes: **Jhinuk Chatterjee

---

## [Author Response · Author response to Decision Letter 0]

9 Sep 2024

Reviewer #1: 

1.Author used the term ‘V-apo’ in many places of this manuscript. What does ‘V-apo’ denotes?

Ans: apo means unbound state of a protein, meaning the vaccine is in a state without binding to any receptors [1], We have revised it in the manuscript.

2.“Codon optimization for Escherichia. coli synthesis allowed…”- here the part ‘Escherichia. coli’ should be written as ‘Escherichia coli’. Their should be no full stop between Escherichia and coli

Ans: We have corrected it in the manuscript.

3.“Among the Paramyxoviridae family of viruses is the genus Henipavirus”- in the sentence the family and genus should be italicized.

Ans: We have corrected it in the manuscript.

4.“The P gene's co transcriptional mRNA editing produces V and W proteins…”- what does the P, V and W indicate?

Ans: The P gene means Phosphoprotein gene but the V and W proteins are not abbreviations but rather designations for specific proteins encoded by P gene [2]. We have corrected it in the manuscript.

5.“… attract viral glycoproteins, viral RNPs…”- what does the RNPs mean?

Ans: RNPs stands for Ribonucleoproteins. The RNP is the primary unit responsible for the synthesis of vRNA, and as a result, it plays a significant role in the virus life cycle [3, 4]. We have corrected it in the manuscript.

6.In the manuscript, you can use the term ‘linear BCL’ instead of ‘LBL’, as the ‘LBL’ also represents the term ‘lymphoblastic lymphoma’.

Ans: We have corrected this term according to your instruction throughout the revised manuscript.

7.“Reverse vaccinology involves identifying antigenic…”- correct the grammar (“involves in identifying”).

Ans: We have corrected this grammatical mistake.

8.Italicized the words: in-silico, in-vitro, in-vivo.

Ans: We have corrected this to your instruction.

9.On the basis of which parameters, these proteins were selected, like- homologous with human & mice proteins, antigenicity, allergenicity, toxicity, etc. Explain.

Ans. Mainly the proteins G, M and F were selected based on pathogenesis. These three proteins were efficient in binding and attaching with host cells. If a vaccine is designed targeting these proteins, the virus will never be able to attach to host cells and ultimately the chance of infection will be limited then as virus require specific attachment with host cells. Without attachment with host cells, virus is not capable of causing infections. The antigenicity, allergenicity, toxicity were also a basis of selection in predicting the individual epitopes where antigenic, non-allergenic and non-toxic epitopes were selected to construct the vaccine.

10.“The full amino acid sequences of the Henipavirus hendraense (HeV)…”- italicized the part ‘Henipavirus hendraense’.

Ans: We have italicized this based on your instruction.

11.In the CTL epitopes prediction, mention the parameters with input values, i.e. set here, like- peptide length, selected alleles, threshold for strong binders: % Rank, threshold for weak binders: % Rank, etc. And why not you use the other well-known alternative- NetCTL 1.2? Is there any specific reason or advantages using NetMHC-4.0 and IEDB server? How did you combine the results from both servers in this study?

Ans. The peptide length was 9, selected alleles were HLA-A*01:01, HLA-A*02:01, HLA-A*02:03, HLA-A*02:06, HLA-A*03:01, HLA-A*11:01, HLA-A*23:01, HLA-A*24:02, HLA-A*26:01, HLA-A*30:01, HLA-A*30:02, HLA-A*31:01, HLA-A*32:01, HLA-A*33:01, HLA-A*68:01, HLA-A*68:02, HLA-B*07:02, HLA-B*08:01, HLA-B*15:01, HLA-B*35:01, HLA-B*40:01, HLA-B*44:02, HLA-B*44:03, HLA-B*51:01, HLA-B*53:01, HLA-B*57:01, HLA-B*58:01, strong binder percentile rank was ≦1.00. IEDB and NetMHC-4.0 provide better output result with higher feasibility and accuracy than NetCTL 1.2. We combined the result by matching the peptides from these two servers.

12.What threshold value was set to compute the antigenicity in VaxiJen 2.0 server?

Ans. The threshold value was 0.4.

13.Mention the input parameters i.e. used to predict the HTL and linear BCL epitopes.

14.“… their respective databases to learn about the allergenicity, toxicity, and antigenic properties of the selected epitopes.”- the ‘allergenicity’, ‘toxicity’ and ‘antigenic properties’ are written in wrong sequence. Please check again and correct it

Ans. We have corrected the sequence.

15.Mention the selected areas or regions and alleles used in population coverage analysis

Ans. The selected regions were the following: East Asia, Northeast Asia, South Asia, Southeast Asia, Southwest Asia, Europe, East Africa, West Africa, Central Africa, North Africa, South Africa, West Indies, North America, South America, Central America, and Ocenia. The MHC-1 alleles that were used were HLA-A*01:01, HLA-A*02:01, HLA-A*02:03, HLA-A*02:06, HLA-A*03:01, HLA-A*11:01, HLA-A*23:01, HLA-A*24:02, HLA-A*26:01, HLA-A*30:01, HLA-A*30:02, HLA-A*31:01, HLA-A*32:01, HLA-A*33:01, HLA-A*68:01, HLA-A*68:02, HLA-B*07:02, HLA-B*08:01, HLA-B*15:01, HLA-B*35:01, HLA-B*40:01, HLA-B*44:02, HLA-B*44:03, HLA-B*51:01, HLA-B*53:01, HLA-B*57:01, HLA-B*58:01. The MHC-2 alleles that were used were HLA-DRB1*01:01, HLA-DRB1*03:01, HLA-DRB1*04:01, HLA-DRB1*04:05, HLA-DRB1*07:01, HLA-DRB1*08:02, HLA-DRB1*09:01, HLA-DRB1*11:01, HLA-DRB1*12:01, HLA-DRB1*13:02, HLA-DRB1*15:01, HLA-DRB3*01:01, HLA-DRB3*02:02, HLA-DRB4*01:01, HLA-DRB5*01:01, HLA-DQA1*05:01, HLA-DQB1*02:01, HLA-DQA1*05:01, HLA-DQB1*03:01, HLA-DQA1*03:01, HLA-DQB1*03:02, HLA-DQA1*04:01, HLA-DQB1*04:02, HLA-DQA1*01:01, HLA-DQB1*05:01, HLA-DQA1*01:02, HLA-DQB1*06:02, HLA-DPA1*02:01, HLA-DPB1*01:01, HLA-DPA1*01:03, HLA-DPB1*02:01, HLA-DPA1*01:03, HLA-DPB1*04:01, HLA-DPA1*03:01, HLA-DPB1*04:02, HLA-DPA1*02:01, HLA-DPB1*05:01, HLA-DPA1*02:01, HLA-DPB1*14:01

16.Why did you use the such linkers ‘EAAAK, AYY, AK, KFER’ to join the epitopes. What are the significances of them? Please explain.

Ans. Linkers are used to separate epitopes, with the goal of ensuring that the features of flexibility, cuttability, and solidity of the epitopes are not disrupted in any way. The immunogenicity of an epitope vaccination can be improved by using AYY and EAAAK linkers [5]. In vaccines, the AK linker can keep HTL epitopes active independently of the immune system [6]. The linear BCL epitopes can be effectively linked using the KFER linker. EAAAK is a rigid peptide linker that forms α-helixes and has a closed-packed backbone, allowing for intramolecular hydrogen bonding. In contrast to flexible linkers, rigid linkers offer numerous advantages. By maintaining a constant distance between the epitopes with minimal interference, EAAAK linkers offer an efficient separation of the functional domains, thereby preserving their individual functional properties [7]. We have added the reasons on the revised manuscript.

17.“Two more sites (http://scratch.proteomics.ics.uci.edu (49) and http://www.ddg-pharmfac.net/va xijen/VaxiJen/VaxiJen.html) (37), were used to verify the antigenicity of the vaccine construct”- write the proper names of the web-servers before the URLs.

Ans. We have corrected it.

18.“We identified potential cysteine disulfide bonding pairs within the vaccine structure using the Disulfide by Design 2.12 algorithm…”- Check the version. It may be ‘2.13’.

Ans. We have corrected it.

19.Mention the input parameters used in ElliPro server.

Ans. In the epitope prediction parameters for ElliPro, the minimum score and the maximum distance were set to 0.5 and 6, respectively. We have added it to the revised manuscript.

20.In the Cluspro 2.0 server, information should be provided which chain authors have used: single chain or both chains in case of TLRs?

Ans. We have corrected it on the methodology section of docking. We blanked the chain id on Cluspro 2.0 to use both chains.

21.Why did you use PDBsum with PyMOL? Is there any necessity?

Ans. PDBsum is efficient at providing interactions of salt bridges, H2 bonds etc between vaccine and TLRs [8]. The three-dimensional (3D) visualization of proteins, nucleic acids, small molecules, electron densities, surfaces, and trajectories has been extensively used with PyMOL, a cross-platform molecular graphics tool. It is also capable of modifying molecules, ray tracing, and producing movies [9].

22.“We used the MM-GBSA methods, which rely on molecular mechanics and the Generalized Born method…”- what does this ‘Generalized Born method’ mean?

Ans. The solvation free energy of molecules is estimated using the Generalized Born (GB) method, a computational chemistry approximation that simplifies the complex interactions between a molecule and its solvent, which is typically water. The GB method represents the solvent as a continuous medium, as opposed to individual solvent molecules. It is based on the Born equation, which identifies the electrostatic solvation energy of a charged sphere in a dielectric medium. The method applies this concept to complex molecular structures by calculating the effective Born radii of each atom in the molecule. This value is indicative of the depth to which each atom is embedded and its interaction with the solvent. Molecular dynamics simulations frequently utilize the GB method in conjunction with Molecular Mechanics (MM) in MM-GBSA calculations to estimate binding free energies, as it strikes a balance between computational efficiency and accuracy [10].

23.“This server utilizing the Escherichia. Coli: K12 codon framework can assess the protein's expression level by…”- write ‘Escherichia. Coli’ as ‘Escherichia coli’.

Ans. We have corrected it.

24.“For the purpose to clone the optimized vaccine gene sequence into the E. coli plasmid vector pET-28a(+)”- italicize the part ‘E. coli’ as ‘E. coli’.

Ans. We have italicized this scientific name.

25.Why did you select this pET-28a(+) vector? Is there any significance to choose these restriction sites (Ecp53kI and PshAI)?

26.“… a regimen consisting of three doses was established…”- why three doses were required? What was the time duration during this study for immune simulation? In which time points the mentioned three doses were introduced?

27.“… and the duration of the steps (one, eighty-four, and one hundred eighty-six), were initially assigned to their respective default values.”- how did author calculate these values?

Ans. 3 vaccines doses are given at 28 days interval where 1 time-step = 8 hours = Day 1, 84 time-steps = 672 hours = Day 28, 168 time-steps = 1344 hours = Day 56 

28.The images’ quality is/are very poor in resolution. Author should provide high resolution images.

29.In the table 1, mention the start and end points of the CTL peptides, also mention the alleles used here.

Ans. Everything is included on the table 1 and the alleles are also mentioned on the revised manuscript.

30.In the table 2, mention the start and end points of the HTL peptides.

Ans. Everything is included on the table 2 of the revised manuscript.

31.In the table 3, mention the end points of the BCL peptides.

Ans. It is mentioned now in the table 3.

32.In the population coverage analysis, why did author selected only specific countries, and why not world population instead of these countries? And what are the alleles considered here?

Ans. We did not select specific countries. We selected regions East Asia, Northeast Asia, South Asia, Southeast Asia, Southwest Asia, Europe, East Africa, West Africa, Central Africa, North Africa, South Africa, West Indies, North America, South America, Central America, Ocenia to represent larger population. We have newly added the coverage all over the world. The MHC-1 alleles that were chosen were HLA-A*01:01, HLA-A*02:01, HLA-A*02:03, HLA-A*02:06, HLA-A*03:01, HLA-A*11:01, HLA-A*23:01, HLA-A*24:02, HLA-A*26:01, HLA-A*30:01, HLA-A*30:02, HLA-A*31:01, HLA-A*32:01, HLA-A*33:01, HLA-A*68:01, HLA-A*68:02, HLA-B*07:02, HLA-B*08:01, HLA-B*15:01, HLA-B*35:01, HLA-B*40:01, HLA-B*44:02, HLA-B*44:03, HLA-B*51:01, HLA-B*53:01, HLA-B*57:01, HLA-B*58:01. 

The MHC-2 alleles that were used were HLA-DRB1*01:01, HLA-DRB1*03:01, HLA-DRB1*04:01, HLA-DRB1*04:05, HLA-DRB1*07:01, HLA-DRB1*08:02, HLA-DRB1*09:01, HLA-DRB1*11:01, HLA-DRB1*12:01, HLA-DRB1*13:02, HLA-DRB1*15:01, HLA-DRB3*01:01, HLA-DRB3*02:02, HLA-DRB4*01:01, HLA-DRB5*01:01, HLA-DQA1*05:01, HLA-DQB1*02:01, HLA-DQA1*05:01, HLA-DQB1*03:01, HLA-DQA1*03:01, HLA-DQB1*03:02, HLA-DQA1*04:01, HLA-DQB1*04:02, HLA-DQA1*01:01, HLA-DQB1*05:01, HLA-DQA1*01:02, HLA-DQB1*06:02, HLA-DPA1*02:01, HLA-DPB1*01:01, HLA-DPA1*01:03, HLA-DPB1*02:01, HLA-DPA1*01:03, HLA-DPB1*04:01, HLA-DPA1*03:01, HLA-DPB1*04:02, HLA-DPA1*02:01, HLA-DPB1*05:01, HLA-DPA1*02:01, HLA-DPB1*14:01

33.What was the basis for selecting the adjuvant ‘50S ribosomal protein L7/L12’?

Ans. The efficacy of vaccination and the maturation of dendritic cells are enhanced by the addition of Large ribosomal subunit protein bL12 OS=Mycobacterium tuberculosis adjuvants, as demonstrated in laboratory trials [11]. That is why this was selected as an adjuvant.

34.“The SOLpro server suggested that the vaccine had the potential to dissolve effectively when expressed in E. coli, as evidenced by its score of 0859720.”- recheck the value, it may be ‘0.859720’.

 Ans: We have corrected the value.

35.“The SOPMA server predicted that the vaccine's 2º structure would 369 consist of a random coil (23.68%), extended strands (18.22%), and an alpha helix (53.64%) (S1 Table, Figs 4B and 4C).”- re-write the S1 Table as ‘Table 5’ and change accordingly (as you put the same table in supporting documents also).

Ans. It is fixed in case of Table 5 as well as for the supplementary tables.

36.“Conversely, the preliminary model's score of -2.69 undoubtedly suggests that the projected model is more accurate (Fig 5C).”- explain in detail, in what basis you had concluded that the latest refined model was better.

Ans. A Z-score is frequently employed in computational biology, particularly in structural bioinformatics, to evaluate the integrity of a model structure, such as a protein model. The Z-score is a measure of the extent to which the score of a specific model (e.g., energy or other structural criteria) deviates from the mean score of a set of reference models, typically expressed in standard deviations. In general, a negative Z-score suggests that the model has a lower energy than the average of the reference set [12].

37.In the immune simulation, some graphs are missing, i.e. Antigen and immunoglobulins, CD8 T-cytotoxic lymphocytes, Cytokines. These graphs are important to describe the immune responses, please add these in the manuscript and discuss their results.

Ans. We have corrected it.

38.“Vaccines against this virus were developed using RV and …”- use the full form of ‘RV’ at least once in the manuscript, like- ‘reverse vaccinology (RV)’.

Ans. We have corrected the it.

39.“… as evidenced by its aliphatic index of 8211…”- re-check the value, it may be ‘82.11’.

Ans. We have corrected the value.

40.“Calculated to be -3.44 KJ/mol, the Z-score further indicates the vaccine model's overall quality.”- how does the Z score indicate the overall quality? Explain.

Ans. A protein's Z-score is the energy separation between the native fold and the average of an ensemble of misfolds, expressed in units of the standard deviation of the ensemble. Z-scores are frequently implemented to evaluate the capacity of knowledge-based potentials to distinguish the native fold from other alternatives [13]. Positive Z-scores typically indicate inaccurate or problematic components of a model [14]. We have included shortly in the discussion section.

41.“Ahmad, F, Albutti A, Tariq, et al. 2022 designed an antiviral drug targeting the G protein of HeV (93).”- re-write this as “Ahmad F et al. 2022 designed…”. “Kamthania, M., Srivastava, S., Desai, M. et al. 2019 designed a HeV vaccine …” as “Kamthania M. et al. 2019 designed …”.

Ans: We have corrected it according to your suggestion.

42.“Because it is a preventative treatment that encourages the immune system to work naturally against HeV, our tailored therapy is better than their study.”- Kindly explain in what way you had concluded this statement in favour of your study.

Ans. We have corrected it.

Reviewer #2: 

Comment 1: Introduction section is not appealing and a substantial analysis with validated approaches are crucial. For example, what question you

---

## [Decision Letter · Decision Letter 1]

27 Sep 2024

PONE-D-24-26158R1Designing novel multiepitope mRNA Vaccine Targeting Hendra Virus (HeV): An Integrative Approach Utilizing Immunoinformatics, Reverse Vaccinology, and Molecular Dynamics SimulationPLOS ONE

Dear Dr. Hossain,

Thank you for submitting your manuscript to PLOS ONE. After careful consideration, we feel that it has merit but does not fully meet PLOS ONE’s publication criteria as it currently stands. Therefore, we invite you to submit a revised version of the manuscript that addresses the points raised during the review process.

We look forward to receiving your revised manuscript.

Kind regards,

Rajesh Kumar Pathak, Ph.D.

Academic Editor

PLOS ONE

Journal Requirements:

**Additional Editor Comments:**

Please include a brief justification for the use of mRNA vaccines in both the discussion and conclusion to address the reviewer’s suggestion. Additionally, it is recommended to change the background of Figure 6 from black to white for better quality.

Reviewers' comments:

Reviewer's Responses to Questions

**Comments to the Author**

1. If the authors have adequately addressed your comments raised in a previous round of review and you feel that this manuscript is now acceptable for publication, you may indicate that here to bypass the “Comments to the Author” section, enter your conflict of interest statement in the “Confidential to Editor” section, and submit your "Accept" recommendation.

Reviewer #1: All comments have been addressed

Reviewer #2: All comments have been addressed

Reviewer #3: All comments have been addressed

2. Is the manuscript technically sound, and do the data support the conclusions?

Reviewer #1: Yes

Reviewer #2: Yes

Reviewer #3: Yes

3. Has the statistical analysis been performed appropriately and rigorously? 

Reviewer #1: Yes

Reviewer #2: Yes

Reviewer #3: Yes

4. Have the authors made all data underlying the findings in their manuscript fully available?

Reviewer #1: Yes

Reviewer #2: Yes

Reviewer #3: Yes

5. Is the manuscript presented in an intelligible fashion and written in standard English?

Reviewer #1: Yes

Reviewer #2: Yes

Reviewer #3: Yes

6. Review Comments to the Author

Reviewer #1: (No Response)

Reviewer #2: The manuscript is in the accepted form now. Currently, I have no additional comments for the author.

Reviewer #3: Comments are addressed. Please include justification of mRNA vaccine in your discussion and conclusion.

7. PLOS authors have the option to publish the peer review history of their article (what does this mean?). If published, this will include your full peer review and any attached files.

Reviewer #1: No

Reviewer #2: No

Reviewer #3: No

---

## [Author Response · Author response to Decision Letter 1]

30 Sep 2024

Journal Requirements

Ans. We highly appreciate that you have mentioned this. We have verified each citation and reference and have substituted the retracted references with the appropriate ones. Furthermore, we have adjusted the citation style in the revised manuscript in accordance with the journal's specifications (PLoS). The file titled Revised Manuscript with Track Changes comprises all modifications, which are indicated in yellow.

Additional Editor Comments:

Please include a brief justification for the use of mRNA vaccines in both the discussion and conclusion to address the reviewer’s suggestion. Additionally, it is recommended to change the background of Figure 6 from black to white for better quality.

Ans. We have mentioned the brief justification for the use of mRNA vaccines in both the discussion and conclusion to address the reviewer’s suggestion.

Furthermore, in accordance with your recommendation, we have modified Figure 6. Now, it is displayed on a white background.

Additionally, we have improved the quality of all figures, corrected the table captions, figure captions, and images.

---

## [Editor Report · Decision Letter 2]

4 Oct 2024

Designing novel multiepitope mRNA Vaccine Targeting Hendra Virus (HeV): An Integrative Approach Utilizing Immunoinformatics, Reverse Vaccinology, and Molecular Dynamics Simulation

PONE-D-24-26158R2

Dear Dr. Hossain,

We’re pleased to inform you that your manuscript has been judged scientifically suitable for publication and will be formally accepted for publication once it meets all outstanding technical requirements.

Kind regards,

Rajesh Kumar Pathak, Ph.D.

Academic Editor

PLOS ONE

Additional Editor Comments (optional):

The authors have adequately addressed our concerns, and the manuscript is now acceptable for publication. However, minor corrections are needed in the RMSF images in Figure 8D and 8F, which should be addressed during the proofreading stage.
---

## [Editor Report · Acceptance letter]

11 Oct 2024

PONE-D-24-26158R2 

PLOS ONE

Dear Dr. Hossain, 

I'm pleased to inform you that your manuscript has been deemed suitable for publication in PLOS ONE. Congratulations! Your manuscript is now being handed over to our production team.

Kind regards, 

on behalf of

Dr. Rajesh Kumar Pathak 

Academic Editor

PLOS ONE